# Visualizing the transiently populated closed-state of human HSP90 ATP binding domain

Faustine Henot[1,5], Elisa Rioual[1,2,5], Adrien Favier [1], Pavel Macek[1,3], Elodie Crublet[3], Pierre Josso[2], Bernhard Brutscher[1], Matthias Frech [4], Pierre Gans[1], Claire Loison [2] ✉ & Jerome Boisbouvier [1] ✉

HSP90 are abundant molecular chaperones, assisting the folding of several hundred client proteins, including substrates involved in tumor growth or neurodegenerative diseases. A complex set of large ATP-driven structural changes occurs during HSP90 functional cycle. However, the existence of such structural rearrangements in apo HSP90 has remained unclear. Here, we identify a metastable excited state in the isolated human HSP90α ATP binding domain. We use solution NMR and mutagenesis to characterize structures of both ground and excited states. We demonstrate that in solution the HSP90α ATP binding domain transiently samples a functionally relevant ATP-lid closed state, distant by more than 30 Å from the ground state. NMR relaxation enables to derive information on the kinetics and thermodynamics of this interconversion, while molecular dynamics simulations establish that the ATP-lid in closed conformation is a metastable exited state. The precise description of the dynamics and structures sampled by human HSP90α ATP binding domain provides information for the future design of new therapeutic ligands.

Heat Shock Protein 90 (HSP90) is a ubiquitous ATP-dependant molecular chaperone involved in the folding of a plethora of client proteins, also modulating their cellular activities. HSP90's activity is itself regulated by post-translational modifications such as phosphorylation, nitration, methylation, or acetylation[1] as well as a large number of different co-chaperones[2–7], including Hop, PP5, p23, Sgt1, FKBP51/52, Cyp40, NudC, and Cdc37. During its functional cycle, in order to fold client proteins, the chaperone HSP90 undergoes complex structural rearrangements driven by both ATP binding and hydrolysis[8,9]. Among the identified HSP90-client proteins, a large fraction is related to cancer, such as steroid hormone receptors, tumor suppressor p53, telomerase, hypoxia-inducible factor 1α, and kinases[10]. Furthermore, high cellular HSP90 expression levels are often associated with poor prognoses in many cancer types[11]. Therefore, human HSP90 has been identified as a major anti-cancer drug target. There are four HSP90 homologues in human cells: GRP94, TRAP1, HSP90α, and HSP90β[12].

While GRP94 and TRAP1 are respectively found in endoplasmic reticulum[13] or mitochondria[14], respectively, both HSP90α and HSP90β isoforms are highly abundant in the cytoplasm, representing 1 to 2 % of cellular proteins[15–17]. Under stress conditions or in tumor cells, the level of expression of HSP90 increases up to 7% of the total amount of expressed proteins[10].

HSP90 is a homodimeric chaperone of ca. 170 kDa. Each monomer is composed of 3 domains: C-terminal domain (CTD), middle domain, and N-terminal domain (NTD). The 13 kDa CTD is responsible for HSP90 dimerization, and harbours a MEEVD motif mediating the binding of various co-chaperones with tetratricopeptide repeat (TPR) domains[18–20]. The middle domain (ca. 40 kDa) is involved in binding of both client proteins and co-chaperones and, additionally, modulates hydrolysis of ATP bound to the NTD[3,21,22]. A long charged linker connects the middle domain and the NTD and modulates molecular interactions with client proteins[23]. Finally, the NTD (ca. 25 kDa) is

[1]Univ. Grenoble Alpes, CNRS, CEA, Institut de Biologie Structurale (IBS), 71, avenue des martyrs, F-38044 Grenoble, France. [2]Institut Lumière Matière, University of Lyon, Université Claude Bernard Lyon 1, CNRS, F-69622 Villeurbanne, France. [3]NMR-Bio, 5 place Robert Schuman, F-38025 Grenoble, France. [4]Discovery Technologies, Merck KGaA, Frankfurter Straße 250, 64293 Darmstadt, Germany. [5]These authors contributed equally: Faustine Henot, Elisa Rioual. ✉e-mail: claire.loison@univ-lyon1.fr; jerome.boisbouvier@ibs.fr

implicated in client protein and co-chaperone binding[3]. HSP90-NTD possesses an ATP binding site with an unusual structure named Bergerat fold[24]. The particular environment of the ATP binding site offers the possibility to develop HSP90-specific inhibitors that are not affecting the activity of most other ATP-binding proteins. The HSP90α ATP binding site is the target of most therapeutic ligands developed so far, and more than 300 crystal structures of HSP90α-NTD, bound to different ligands, are available in the PDB. A particular challenge for the design of new inhibitors against this important cancer target is the presence of a highly flexible ATP-lid segment that can cover the nucleotide/drug binding site in HSP90[25,26]. Large changes of the ATP-lid conformation also occur during the functional cycle[27]. Changes in the lid conformational dynamics are linked to chaperone activity[28] and nucleotide binding[27,29–34]. But there is no consensus yet regarding the driving forces modulating the conformational changes during the chaperone cycle[28,35]. It was proposed that the lid acts as a nucleotide-sensitive conformational switch of the molecular chaperone activity[36]. There would be a strong correlation between the chaperone activity and the ATP vs. ADP binding[30,37,38]. In contrast, it was also reported that the transitions between the conformational states and the nucleotide binding/unbinding are mainly thermally driven, with large conformational fluctuations on timescales faster than the rate of ATP hydrolysis[39,40]. Last, Southworth et al. mentioned that rather than being irreversibly determined by nucleotide binding, a conformational equilibrium exists between different states[41].

In this article, we report atomic-resolution structural and dynamics information on HSP90α-NTD ATP-lid in solution. We demonstrate that the ATP-lid segment of this crucial human chaperone, in addition to the well-characterized open state, also transiently populates a closed conformation that requires structural rearrangement of peptide segments over a distance of up to 30 Å. Using solution NMR spectroscopy, we determine atomic-resolution models of the open- and closed-ATP-lid state conformations, as well as derive kinetic and thermodynamic information for this structural rearrangement occurring in solution. Complementary molecular dynamics investigation reveals that the conformation of the closed state is metastable on the microsecond time scale. Our study reveals that the closure of the ATP-lid, observed during the ATP-driven functional cycle of human HSP90α, is already sampled in the apo chaperone without binding of ATP. We anticipate that our results will be important for the future drug design of new inhibitors against this challenging drug target.

## Results

### Conformational variability of HSP90α-NTD

HSP90 has been identified as a major therapeutic target, especially against cancer[2,42,43]. Since the late 90's a large number of competitive inhibitors, targeting the ATP binding site of the HSP90α-NTD has entered clinical trials[44–46]. In the context of these drug development efforts, more than 300 atomic resolution structures of isolated HSP90α-NTD in the presence of various ligands have been determined, using X-ray crystallography. These structures can be separated into 8 main groups (Fig. 1a–c) showing distinct conformational properties. Pairwise superimposition of the HSP90α-NTD centroid structures representing each cluster reveals that the main structural differences are located on the segment covering the nucleotide/drug binding site. While large parts of these 8 centroids superimpose with an average root-mean-square deviation (RMSD) of ca. 0.2 Å, the ATP-lid[25] (defined here from residues M98 to V136 including three helical segments) shows higher heterogeneity with an average pairwise RMSD of 1.4 Å (Fig. 1d). In the six available apo structures (from which five belong to cluster 3 and one to cluster 2), and all nucleotide- or ligand-bound structures of the isolated HSP90α-NTD, the ATP-lid is in the so-called open state[33] and does not cover the ATPase site and corresponding drug binding site. The observed structural variability of HSP90α-NTD structures indicates that the ATP-lid, in the open state, can adopt

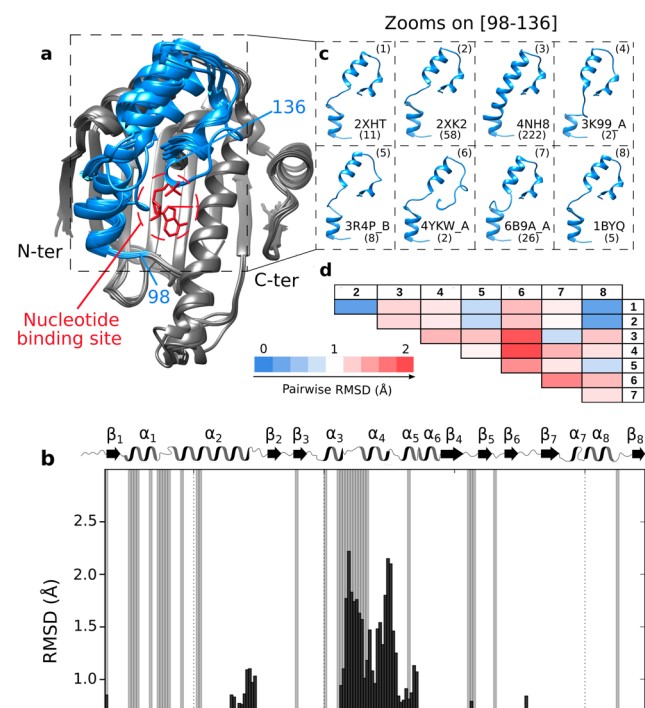

**Fig. 1 | Analysis of available structures of human HSP90α-NTD.**
**a** Superimposition of 8 centroids representing the 8 clusters describing the 334 structures of isolated HSP90α-NTD available in the Protein Data Bank (on January the 5th of 2021). Clustering was performed using the MaxCluster program (http://www.sbg.bio.ic.ac.uk/maxcluster) with an Average Linkage type of hierarchical clustering and a threshold value of 1.05. In blue is depicted the segment [98–136] and in red the nucleotide. **b** Histogram of the averaged pairwise RMSD between Cα backbone atoms of the 8 centroids in Å as a function of the residue number (black). Grey bars represent non-assigned backbone residues[49]. On top of the histogram: secondary structure elements as a function of the residue number (α for helices and β for sheets). **c** Zooms on the segment [98–136], that shows the highest structural variability, for all the 8 centroids superimposed using the Chimera MatchMaker command. PDB ID and a number of structures present in each cluster are disclosed next to each centroid. **d** Table representing pairwise RMSD in Å between Cα backbone atoms belonging to the segment [98–136] of each pair of the 8 centroids superimposed using Chimera MatchMaker command (PDB IDs: 1: 2XHT, 2: 2XK2, 3: 4NH8, 4: 3K99_A, 5: 3R4P_B, 6: 4YKW_A, 7: 6B9A_A, 8: 1BYQ). Going from blue to red the RMSD values increase. Centroids from clusters (1), (2), and (8) are highly similar (low pairwise RMSD values) on the segment [98–136], but these three centroids differ mainly on another segment of the protein [64–75].

various conformations in a crystalline environment depending on the crystallization conditions, and the stabilization by ligands or additives. However, a detailed picture of the conformational properties of the ATP-lid segment in solution is of utmost importance for future drug design, as this segment is located next to the major drug-binding site of human HSP90. Therefore, we initiated a structural and dynamic investigation of HSP90α-NTD using solution NMR spectroscopy to characterize at atomic resolution the conformations sampled by the ATP-lid.

### The ATP-lid of HSP90α-NTD populates two distinct conformations in solution

Previous NMR investigations reported that human HSP90α-NTD amide backbone NMR signals from N105 to K116 in the ATP-lid segment were undetectable[47–49] (Fig. 1b). This observation suggests that

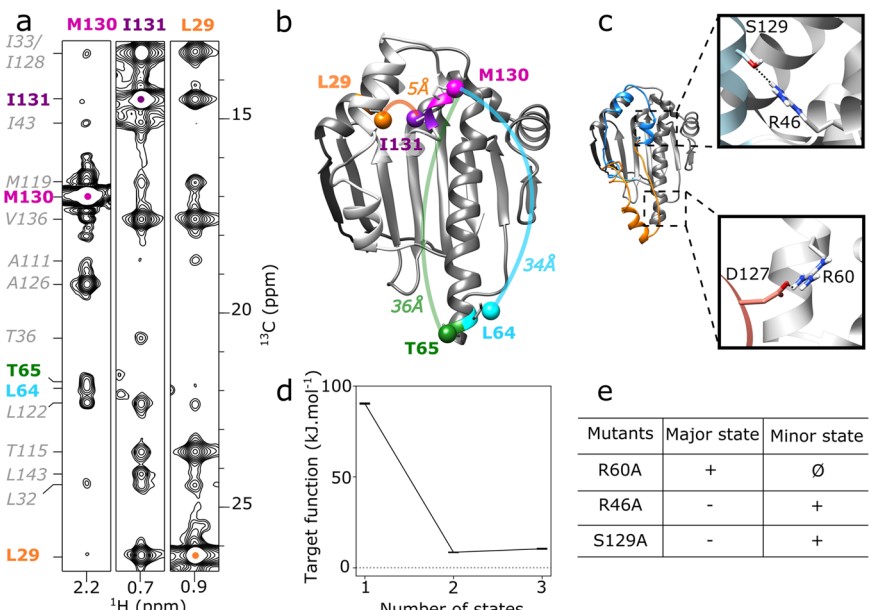

**Fig. 2 | HSP90α-NTD samples two states in solution. a** 2D-strips extracted from $^{13}CH_3$-edited 3D NOESY, at $^{13}C$ frequencies of M130, I131 and L29 methyl groups of WT-HSP90α-NTD. The 3D HMQC-NOESY-HMQC experiment was acquired at 25 °C on a spectrometer operating at a $^1H$ frequency of 950 MHz. **b** Representation on a 3D structure of HSP90α-NTD (PDB 1YES) of examples of intermethyl NOEs involving M130 and I131 with methyl groups distant by more than 30 Å. **c** Representatives two states of HSP90α-NTD calculated simultaneously[74] to satisfy all NMR restraints acquired on WT-HSP90α-NTD. Boxes correspond to zooms on structure displaying interactions stabilizing the conformation of the ATP-lid either in closed (orange) or open conformation (blue) **d** Variation of average CYANA target function for the 20 best structures according to the number of states used to calculate each conformer. **e** Table summarizing intensity increases (+), decreases (−), or disappearance (Ø) of characteristic NOEs corresponding to HSP90α-NTD ATP-lid open and closed states (Supplementary Table 2).

the ATP-lid is undergoing conformational exchange on the micro- to millisecond time scale. The broadening of backbone NMR signals in the ATP-lid is also particularly detrimental for NMR-distance restraint-based structure determination using uniformly $^{13}C,^{15}N$-labeled HSP90α-NTD sample. In order to overcome this problem, we decided to produce perdeuterated HSP90α-NTD, specifically $^{13}CH_3$-labeled on the $A^β$, $I^{δ1}$, $L^{δ2}$, $M^ε$, $T^γ$ and $V^{γ2}$ methyl positions[50]. The resulting 87 $^1H,^{13}C$-labeled methyl groups all gave rise to detectable NMR signals. Most interestingly, 5 of these methyl groups (L107, T109, I110, A111 and T115) belong to the previously unassigned stretch of residues in the ATP-lid (Supplementary Fig. 1). Unambiguous sequence-specific $^1H$ and $^{13}C$ resonance assignments of all detected methyl groups was obtained from a set of through-bond correlation experiments[49], aided by an extensive mutagenesis-driven assignment of 33 methyl probes[51] (Supplementary Fig. 2).

Next, we recorded methyl NOESY spectra to derive a set of inter-methyl distance restraints. The applied labeling scheme that yields protonation only on the methyl moieties at the extremity of hydrophobic side chains is particularly useful to extract long-range distance restraints between methyl groups that are separated by up to 10 Å[52]. For HSP90α-NTD, 597 inter-methyl NOEs (Supplementary Fig. 3a and Supplementary Table 1) could be detected, from which 111 distance restraints were derived that involve at least one methyl group belonging to the ATP-lid segment. All these structurally meaningful restraints were assigned exclusively based on chemical shifts. The inter-methyl distance restraints were complemented by 41 short-range backbone $H_N$-$H_N$ distance restraints (Supplementary Fig. 3a and Supplementary Table 1) derived from a 3D $^{15}N$-edited NOESY spectrum, and 54 backbone φ,ψ-dihedral restraints derived from characteristic backbone chemical shifts[49,53].

Based on this set of experimentally derived structural restraints, we have performed structure calculations using a simulated annealing protocol in torsion angle space[54] to determine the structure and position of the ATP-lid segment relative to the rigid HSP90α-NTD scaffold defined by the two segments [11−97] and [137−223]. To our surprise, the calculated structures showed a large number of distance violations (25) and a high final target energy (in average 90.32 kJ.mol⁻¹). Careful analysis of the reported distance violations revealed a number of NMR-derived restraints that are incompatible with a single structure (Fig. 1a). As an example, a pair of inter-methyl NOEs is detected between the methyl group of I131-$δ_1$ located in the ATP-lid and the methyl moiety of L29-$δ_2$ in helix−1, while another set of NOE correlation peaks indicate that the preceding residue M130-ε in the ATP-lid is spatially close to L64-$δ_2$ and T65-γ, two methyl groups located at the end of the long helix-2 (Fig. 2a, b). However, according to the previously determined X-ray structures of HSP90-NTD (Fig. 1), L29 is more than 30 Å away from L64 and T65. The specific assignment of these methyl groups, the analysis of the through-bond sequence-specific assignments of backbone nuclei and the transfer to methyl-moieties were double checked[49] and completed by an extensive mutagenesis-driven assignment of 23 additional key methyl probes[51] (Supplementary Fig. 2) allowing us to exclude misassignment of these NOEs. As the NMR-derived distance restraints could not be satisfied with a single structure, we made the hypothesis that the ATP-lid populates several conformations. Therefore, an additional round of structure calculations was performed[55] assuming two co-existing conformations of the ATP-lid segment (Fig. 2c), and allowing each experimental distance restraint to be satisfied by either one of the two structures, or both. The introduction of two distinct conformational states in our calculation protocol allowed to satisfy all experimental restraints, and also decreased the final target energy (Fig. 2d) by a factor of 11. Additional calculations, including a third state did not result in further improvement of the target energy (Fig. 2d). To assess in an automated unbiased manner how many protein states are required to satisfy all the experimental distance restraints, we used structural correlation measure that determines the optimal number of states for multi-state structure calculation that can give more clear-cut results than the conventional target-function-based analysis[56]. The structural correlations value is expected to be maximum when the number of states used for the calculation reaches the number of states sampled by the

target protein. The computed structural correlations value is 0.62 for a two states model, and drops to 0.39 for a 3-states model, indicating that a model including two states is sufficient to satisfy all experimental data.

When the two final average structures are superimposed on the common peptide segments [11–97] and [137–223], the backbone RMSD of the ATP-lid segment is ca. 20 Å (Fig. 2c). The first state corresponds to the structure with the ATP-lid in an open position, similar to the bundle of representative structures previously resolved for the isolated HSP90α-NTD (Fig. 1a). The second state shows the ATP-lid in a closed conformation, covering the entire ATP-binding site, and helix−4 located near the end of long helix−2. A careful analysis of these structures showed that 22 NOE-derived distance restraints are satisfied only by the ATP-lid open state, while 5 distance restraints are specific to the ATP-lid closed state (Supplementary Table 2), and the remaining 125 restraints are satisfied in both states.

## Stabilization of ATP-lid closed and open states by mutagenesis

Our structural model of the ATP-lid closed state points toward a salt bridge involving residues R60 and D127 as a potential contributor to the enthalpic stabilization of this so far unknown conformation of HSP90α-NTD (Fig. 2c). Therefore, we speculated that HSP90α-NTD mutations suppressing the R60/D127 electrostatic interaction would push the population equilibrium toward the ATP-lid open conformation. Two $^{13}CH_3$-labeled samples of single-point HSP90α-NTD mutants (D127A and R60A) were prepared and analyzed by NMR. While the D127A mutant showed NMR spectral signatures characteristic of a structurally heterogeneous sample, the R60A mutant yielded a single set of $^1$H-$^{13}$C correlation peaks -similar to the WT (Supplementary Fig. 4a). Analysis of a $^{13}CH_3$-edited NOESY spectrum recorded for this R60A mutant revealed that all expected NOE correlation peaks specific of the ATP-lid closed state were no longer detected for this mutant (Supplementary Fig. 5). This result indicates that disruption of the R60/D127 salt bridge decreases the population of the ATP-lid closed state to a level that is no longer detected by NOESY-type NMR experiments.

In a similar way, our structural model of the ATP-lid open state allowed us to identify a stabilizing hydrogen bond between the side chains of R46 and S129 (Fig. 2c). Again, we produced $^{13}CH_3$-labeled samples of single-point HSP90α-NTD mutants (R46A and S129A), in order to destabilize the ATP-lid open state and increase the population of the closed state. For both mutants, homogeneous $^1$H-$^{13}$C correlation spectra were obtained (Supplementary Fig. 4b), and methyl NOEs characteristic of both the ATP-lid open and closed states were detected. Still, these mutations resulted in a shift of population from the ATP-lid open to closed state, as the intensities of NOE correlation peaks specific for the ATP-lid closed state were enhanced by a factor of ca. 2.8 with respect to those characteristic of the ATP-lid open conformation (Supplementary Fig. 5). The observation of population shifts induced by single-point mutations derived from our structural models of the ATP-lid closed and open states also strongly supports our 2-states model and the existence of an ATP-lid closed state in WT-HSP90α-NTD in solution.

## NMR refinement of structural models for the ATP-lid open and closed states

Since the HSP90α-NTD mutants possess populations of the two ATP-lid conformations strongly skewed toward either an open or a closed state (Fig. 2e), they provide an opportunity to further refine structural models. For structure calculation of the ATP-lid open state, a set of 114 distance restraints, involving at least one methyl group of the ATP-lid segment, was derived from the 3D $^{13}CH_3$-edited NOESY spectrum acquired using $^{13}CH_3$-labeled sample of R60A-HSP90α-NTD. As the experimental set of distance restraints is self-consistent, calculations were performed assuming a single ATP-lid open state for each calculated conformer. No violation of experimental restraints was observed

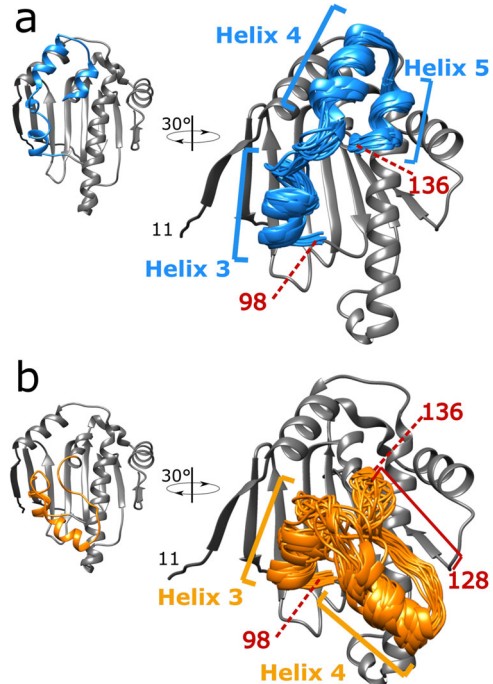

**Fig. 3 | Solution structure ensembles of HSP90α-NTD ATP-lid open and closed states.** For each state, the 20 best CYANA conformers were selected for further restrained molecular dynamics refinement in explicit water. For each panel, the centroid representative conformer of the ensemble is presented on the left in an orientation similar to Fig. 2b, c. On the right the structure ensemble is tilted by 30° and was superimposed on the coordinate of the centroid conformers. **a** structure ensemble for ATP-lid open/ground-state calculated using NMR structural distance restraints obtained using R60A-HSP90α-NTD sample. **b** Structure ensemble for ATP-lid closed/excited-state calculated using NMR structural distance restraints obtained using R46A-HSP90α-NTD sample. Helices 3, 4, and 5 correspond to ATP-lid helices. The positions of residues 98 and 136 are also indicated. The location of the unfolded α$_5$ helix in the closed state is also indicated (between 128 and 136).

in the 20 best-calculated structures. After a final refinement step using molecular dynamics in explicit water and in the presence of experimental restraints[57], the structures of the ATP-lid segment can be superimposed on backbone atoms with a RMSD to the average structure of 0.83 Å (Fig. 3a, Supplementary Table 3). The calculated ATP-lid open conformation forms three helices ([100–104], [115–123] and [128–135]) separated by short loop segments. As the backbone signals were not detectable for the segment [105–115], this part of the ATP-lid is less well-defined due to a limited number of restraints. Compared to previously solved structures of HSP90α-NTD (Fig. 1a), the final structure ensemble is closer to cluster 5 ($C_α$ rmsd 1.45 Å) and clusters 1, 2 and 8 ($C_α$ rmsd 1.65 Å in average), while other clusters superimpose with a backbone rmsd higher than 2.3 Å (Fig. 1c, d).

To refine the ATP-lid closed state, we based our structure calculation on NOE-derived distance restraints from the R46A mutant, as this mutation is located far away from ATP-lid in closed state contrary to S129A mutant which may affect ATP-lid structure. For R46A mutant, as NOEs that are specific to the ATP-lid open state can still be detected, although at a reduced intensity, we used the refined structure of the ATP-lid open state to identify and filter-out restraints specific to this conformation (Supplementary Table 2). The remaining 81 inter-methyl NOEs, completed with 36 dihedral and 41 backbone distance restraints (Supplementary Table 1), were used to refine the structure of the ATP-lid segment in the closed state on the rigid HSP90α -NTD scaffold. The final structural ensemble, superimposable to the average structure with a backbone RMSD of 1.5 Å (Fig. 3b), did not show any violation of experimental restraints. The hinge residues allowing the ATP-lid to

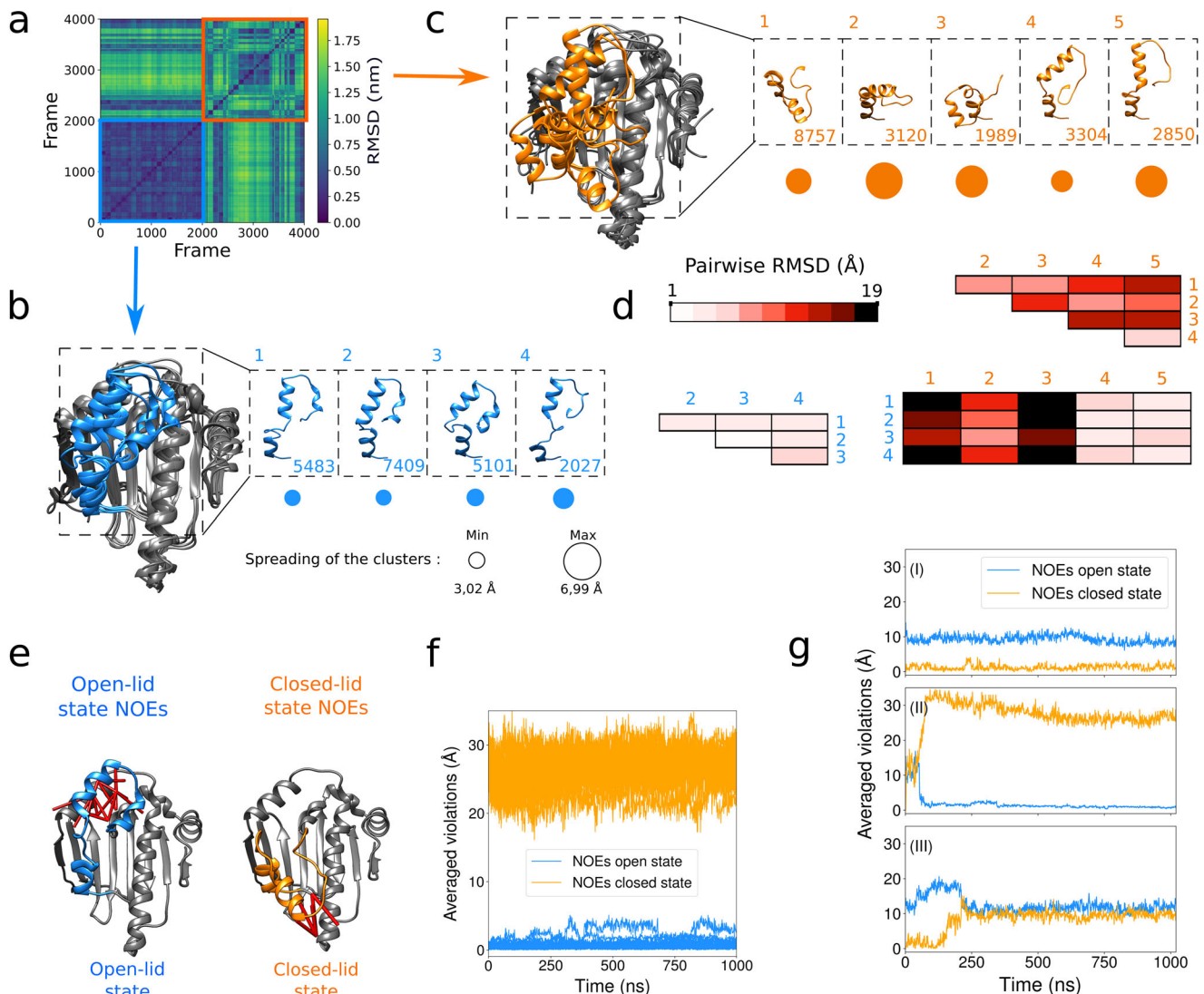

**Fig. 4 | Molecular dynamic investigation of HSP90α-NTD ATP-lid open and closed states. a** Pairwise RMSD among the 4000 ATP-lid structures obtained in the forty Molecular Dynamics simulations of 1 μs. From blue to yellow, the RMSD values increase. The initial models are the two ensembles of experimentally refined structures (see Fig. 3a, b). The first and second half of the 4000 frames are extracted from simulations starting with the ATP-lid in the open state (blue square) or in the closed state (orange square), respectively. **b** Superimposition of the 4 centroids representing the 4 clusters describing 20020 conformations extracted from the simulations starting with ATP-lid open-state conformers. The ATP-lid centroids are represented in b.1, b.2, b.3, and b.4 with the total number of members for each cluster. Below, the circle radii are proportional to the spreading of the cluster (in Å). **c** Same as (b) for the 5 centroids representing the 5 clusters describing 20020 conformations extracted from the simulation starting with ATP-lid closed-state conformers. **d** Pair-wise C$_\alpha$-RMSD between the centroids of the clusters represented in (**b**) and (**c**), with a color-code that is darker when the RMSD is higher. Blue and orange indices correspond to the clusters presented in (**b**) and (**c**) issued from the simulations starting from open and closed states, respectively. **e** Representation of NOE distance restraints characteristics of ATP-lid open and closed states, on the corresponding structures. **f** Violations of characteristics NOEs monitored during the molecular dynamics simulations performed without restraints, for the 20 simulations starting with the ATP-lid in the open state (see methods for definition of computed violations). The values are averaged over all the specific distance restraints depicted in (**e**), either for the ATP-lid open states (blue curves) or for closed states (orange curves) (Supplementary Table 2). **g** As (**f**), for three typical simulations starting with the ATP-lid in the closed state: one remains stable (**I**), one undergoes a transition towards the ATP-lid in the open-state (**II**), and one derives towards region of conformational space that is neither the closed, nor the open ATP-lid state (**III**).

switch from an open to a closed conformation are residues [109–111] and [135–136]. While the helix-3 and helix-4 are conserved in this structure, the segment corresponding to helix−5 in the ATP-lid open state is in an extended conformation in the closed state, thus enabling formation of a salt-bridge between R60 and D127 as well as methyl-methyl contacts between ATP-lid A121, A124, A126, M130 side-chains and methyl moieties L64 and T65.

### The ATP-lid closed state is a metastable excited state
To study the stability of both HSP90 ATP-lid open and closed states, we have explored the conformational landscape around these two

families of conformers using molecular dynamics simulations (without experimental restraints). Representative conformers of the structural ensembles obtained for the ATP-lid open and closed states were used as starting models for forty 1 μs-molecular dynamics simulations. Simulations starting from an ATP-lid in the open state remained stable over the whole 1-μs trajectory. More quantitatively, the pairwise Cα-RMSD over 2000 conformations extracted every 10 ns from the twenty 1-μs long trajectories remains low (Fig. 4a bottom left corner in a blue square), with an average of 4.0 Å. A clustering procedure performed on 20 020 snapshots of the trajectories (every 1 ns) yielded four representative conformations depicted in Fig. 4b, with their respective

population and spreading. As expected, the pairwise RMSD between these four clusters remains low (see Fig. 4d), similar to the RMSD with known X-Ray structures (see Supplementary Table 4). Furthermore, the NOE contacts characteristic of the ATP-lid in the open state (Fig. 4e and the blue curves of Fig. 4f) are respected during the whole simulations, even if no restraints were applied. In contrast, the 2000 conformations extracted from the 20 trajectories starting with the ATP-lid in the closed state showed a larger averaged pairwise RMSD (9.3 Å, see the top right corner of Fig. 4a in an orange square). The five clusters obtained from these simulations and their respective population and spreading are depicted in Fig. 4c. The pairwise RMSD among these clusters, and between these clusters (Fig. 4d), show a structural diversity that may originate either from a complex energy landscape around an excited state, and/or from the uncertainties of the initial experimental structures. According to the Root-Mean-Square-Fluctuations (RMSF) profiles (Supplementary Fig. 6), the increased conformational fluctuations of the ATP-lid in the closed state does not seem to modify the dynamics of the rest of the protein. Analyzing the dynamics of the ATP-lid in individual trajectories reveals three different behaviors that demonstrate that the simulations are not ergodic. The first scenario is represented by 7 trajectories (among 20) that were considered as stable, i.e., they explore conformational space around the starting conformation without large violations of the NOE contacts that are characteristic of ATP-lid in the closed state during the whole 1 μs-simulations (Fig. 4g (I) and Supplementary Fig. 7). This indicates that the refinement procedure has provided a state that is metastable in the microsecond timescale, even in absence of experimental restraints. They are represented in Fig. 4c by clusters 1 to 3. In contrast, among the unstable trajectories, about 9 show a transition toward conformations almost compatible with the NOE restraints characteristic of the ATP-lid open state (Fig. 4g (II)). They are characterized by RMSD values, in the anti-diagonal corners of Fig. 4a, as low as 1.5 Å and correspond to the clusters 4 and 5 in Fig. 4c. Finally, in 20 % of the trajectories, the ATP-lid populates conformations that are neither clearly the ATP-lid closed, nor the open state (Fig. 4g (III)). The observation of several spontaneous transitions of the ATP-lid from the closed to the open state, and the stability of the MD trajectories starting from the open state, indicate that the ATP-lid open state (the only state observed by X-ray crystallography, Fig. 1), is the ground state of HSP90-NTD, while the ATP-lid closed state is a metastable, less populated excited state. This conclusion is also supported by the observation of a small number (5) of characteristic NOEs for the closed ATP-lid state compared to a larger number (22) detected for the open ATP-lid ground state (Supplementary Table 2).

## ATP-lid open and closed states exchange on the millisecond time scale

The fact that a single set of NMR signals is detected for HSP90α-NTD (WT, R60A and R46A mutants) indicates that the ATP-lid open and closed states are in fast exchange with respect to the NMR time scale ($\tau_{ex} < {\sim}10$ ms). The observed severe line broadening of amide backbone resonances in the ATP-lid segment also points toward exchange dynamics on the micro- to millisecond time scale. In order to further quantify the kinetics (and thermodynamics) of the ATP-lid structural rearrangement, NMR relaxation-dispersion experiments were performed at 293 K and two or three different magnetic field strengths using $^{13}C–^1H$ methyl and $^{15}N$ backbone amide probes, respectively (Fig. 5 and Supplementary Fig. 8). These data reveal that conformational exchange is mainly sensed by nuclei in the ATP-lid segment, as well as protein regions that are in close contact with the ATP-lid (Fig. 5a). Particularly strong exchange-contributions were detected in the hinge regions of the ATP-lid segment: residues [T109-T115] and V136. Assuming a simple two states model to describe the structural rearrangement of the ATP-lid, a systematic grid search of the model parameters fitting the relaxation dispersion data detected on

backbone and methyl probes was performed. A single minimum was identified, indicating that the experimental data can be interpreted with the presence of a single low-populated minor state (Supplementary Fig. 8A). A global fit of the $^{13}CH_3$-CPMG relaxation dispersion, characterized by a higher signal-to-noise ratio, enabled us to determine the global exchange rate $k_{ex} = k_1 + k_{-1} = 2490 \pm 61\ s^{-1}$ and relative populations of the 2 states of 96.8 ± 0.1 % and 3.2 ± 0.1 %. This corresponds to half-life times of 8.6 ± 0.4 ms for the ground state, and 0.29 ± 0.01 ms for the excited state. These populations and exchange rates correspond to a difference of Gibbs free energy of 8.3 ± 0.2 kJ.mol$^{-1}$ between the 2 states, with activation energy (from the ground to the transition state) of 61.0 ± 0.2 kJ.mol$^{-1}$[58]. While there is no direct experimental evidence that the excited state detected by the CPMG relaxation-dispersion is indeed the HSP90α-NTD closed state structure determined combining $^{13}CH_3$-NOESY and mutagenesis, the simplest model explaining all the experimental data is the two-state model where the excited state detected by the CPMG relaxation-dispersion is indeed the HSP90α-NTD closed state (Fig. 3b).

A similar strategy was used to assess how R46A- and R60A-mutations modify the kinetics of ATP-lid transition using available $^{13}CH_3$-labeled samples (Supplementary Fig. 9). Global fits of relaxation-dispersion data collected at 288 K revealed an increase in exchange kinetics for R60A ($k_{ex} = 3764 \pm 132\ s^{-1}$) and R46A ($k_{ex} = 4192 \pm 175\ s^{-1}$) compared to WT HSP90α-NTD ($k_{ex} = 2994 \pm 83\ s^{-1}$). However, such fast exchange regimes preclude reliable extraction of state populations[59,60]. Nevertheless, it is interesting to note that destabilization of only the ground ATP lid open state (or the excited ATP-lid closed state) by mutagenesis is expected to increase $k_1$ (respectively $k_{-1}$) and, therefore, the global exchange rate $k_{ex}$, as experimentally observed.

## Discussion

In this work, we have demonstrated that the ATP-lid segment of human HSP90α-NTD samples two distant conformations in solution. We have solved the solution structures of this important anti-cancer drug target with the ATP-lid either in an open or in a closed conformation. Such a closed state with the ATP-lid covering the nucleotide/drug binding site has never been observed before for isolated HSP90α -NTD, neither in the apo state nor in complex with nucleotides or drugs (Fig. 1). In addition to our NMR data, we have used mutagenesis and molecular dynamics simulation to independently validate the existence of this metastable, low populated closed state in human HSP90α -NTD. Acquisition of 3D-NOESY spectra using methyl-labeled samples of HSP90α -NTD in the presence of ADP or a resorcinol inhibitor reveals that in both cases, the NOEs specific of the ATP-lid closed state are observed with intensities similar to HSP90α -NTD apo state (Supplementary Fig. 10), indicating that this minor conformation is preserved in the presence of each one of these two ligands.

Using Molecular Dynamics simulations and NMR relaxation-dispersion experiments, we could establish that the ATP-lid in the open conformation is thermodynamically more stable than the ATP-lid in the closed state, and that the population of the exited (closed) state is only 3–4 % at room temperature (Fig. 5). The ATP-lid open and closed states interconvert at a rate ($k_{ex} = k_1 + k_{-1}$) of 2.5 kHz, which seems to be very fast for such an extensive structural rearrangement. However, structural rearrangements at fast time scales were already reported for other conformational exchange events in proteins. An example is EIN, a 128 kDa homodimeric protein that switches from an open to a closed state at a frequency higher than 10 kHz[61]. Furthermore, PET fluorescence quenching experiments by Schulze et al.[26] monitoring yeast HSP90-NTD ATP-lid structural rearrangement in isolated monomeric constructs reveals a motion of the ATP-lid with a characteristic rate of 1.5 kHz, even though a full closure of the lid over the nucleotide/drug binding site has not been observed. This frequency range is similar to our results obtained by NMR relaxation dispersion experiments for

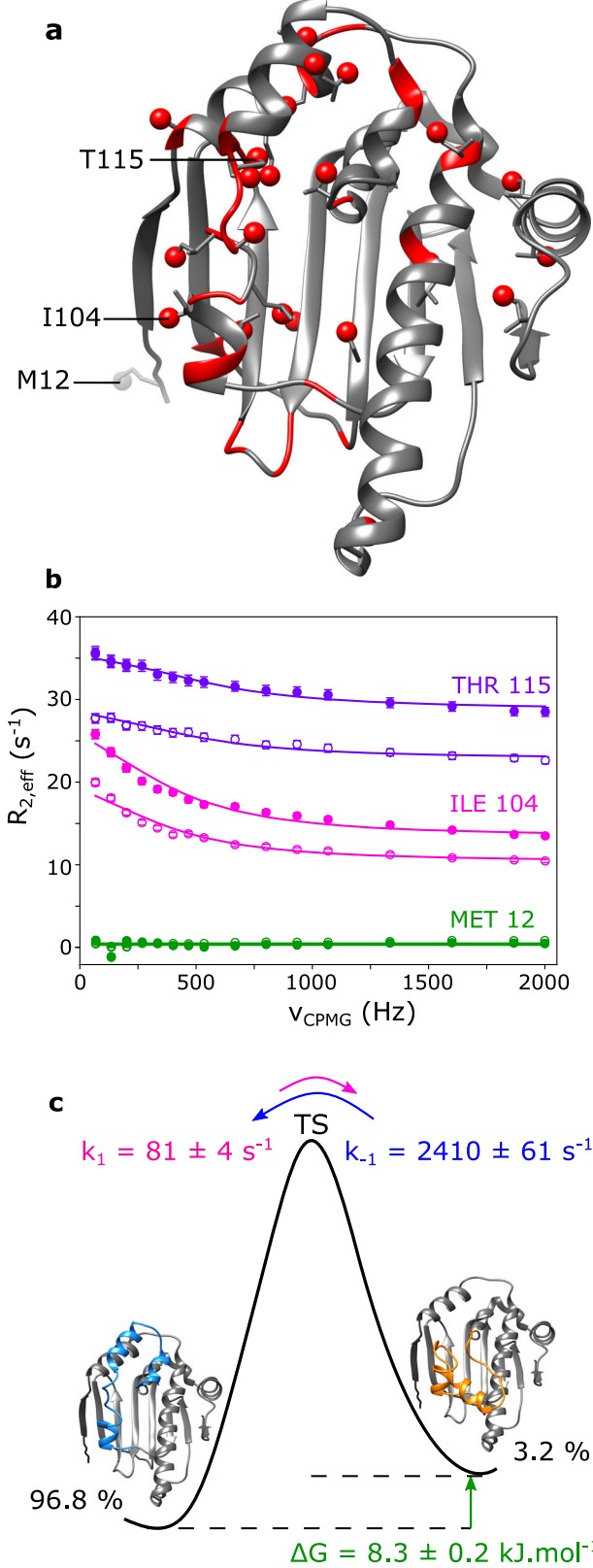

**Fig. 5 | Relaxation dispersion study of HSP90α-NTD. a** 3D structure of human HSP90α-NTD displaying in red methyl- and backbone $^{15}N$- probes for which conformational exchange in the μs-ms time scale was detected. **b** Examples of $^{13}CH_3$ MQ CPMG relaxation dispersion profiles of Thr-115, Ile-104 and Met-12 plotted in purple, pink and green, respectively. The filled circles for each color represent data acquired at 850 MHz, and the empty circles represent data acquired at 700 MHz. Data displayed were acquired at 293 K using a U-[$^2$H, $^{15}$N, $^{12}$C], Ala-[$^{13}C^1H_3$]$^β$, Ile-[$^{13}C^1H_3$]$^{δ1}$, Leu-[$^{13}C^1H_3$]$^{δ2}$, Met-[$^{13}C^1H_3$]$^ε$, Thr-[$^{13}C^1H_3$]$^γ$, Val-[$^{13}C^1H_3$]$^{γ2}$ HSP90-NTD sample. Experimental data were fitted to a two-sites exchange model (global fit of 21 relaxation dispersion curves). Errors for $R_{2,eff}$ rate values were estimated from twice the noise measured in the spectra. However, when errors were less than 2% of the $R_{2,eff}$ rate value, an error of 2% was assumed. **c** Schematic diagram of the energy landscape for the exchange between the ATP-lid open (ground) and closed (excited) states of HSP90α-NTD. Both thermodynamics and kinetics parameters of the exchange, extracted at 293 K, are displayed. Activation energies can be estimated using Eyring equation (assuming that the transmission coefficient κ is 1). From the ground state to the transition state: $ΔG^‡ = 61.0 ± 0.2$ kJ.mol$^{-1}$ and from the excited state to the transition state: $ΔG^‡ = 52.7 ± 0.1$ kJ.mol$^{-1}$. The precision on the data was estimated using Monte Carlo.

structural changes observed for the ATP-lid open-closed transition. The most pronounced rearrangements between the ATP-lid open and closed states are the reorientation of helix-3 and helix-4, as well as the folding/unfolding of helix−5. The MD trajectories describing a full transition of ATP-lid from the closed to the open state show that the ATP-lid can tilt very rapidly, typically in a few hundreds of ns, while the folding of helix-5 and the associated local restructuration were not systematically observed during the 1-μs simulations (Supplementary Fig. 11). Therefore, we may conclude that the transition pathway involves several structural rearrangements occurring at different timescales, with the unwinding/folding of the helix-5 being one of the rate-limiting steps of the transition between ATP-lid open and closed states.

To assist the folding of client proteins, the ATPase HSP90 interacts with a large number of co-chaperones and undergoes a conformational cycle in which a complex set of structural changes occurs. In currently proposed models of the HSP90 functional cycle, the two N-terminal domains of HSP90 transiently dimerize upon binding of ATP, client proteins, and co-chaperones[62]. Dimerization is associated with an exchange of the first β-strand (β1) between both N-terminal domains, and a switch of the ATP-lid from an open to a closed state[25,26,63,64]. Cryo-EM structures of full-length HSP90 co-vitrified with ATP, a co-chaperone and a client protein[65,66] reveal an ATP-lid segment in a closed conformation associated to a dimerization of the N-terminal domains and a swap of the β1-strands between both N-terminal domains (Fig. 6a−c). To investigate whether the closed ATP-lid conformation is responsible for dimerization of HSP90-NTD, we have identified 16 methyl-methyl proximities in the N-terminal domain of full-length human HSP90α (PDB 7L7J) that are characteristic of the dimeric HSP90 with a closed ATP-lid and swapped β1-strands (Supplementary Table 5). However, investigation of the inter-methyl NOEs observed for the HSP90-NTD mutant R46A, showing an enhanced population of the ATP-lid closed-state ensemble, revealed the absence of all the putative NOEs characteristic of dimerization of HSP90α-NTDs, indicating that the ATP-lid closed state of HSP90α-NTD stays monomeric in solution. SEC-MALS analysis of both WT and R46A mutant of HSP90α −NTD confirms that the protein is monomeric in solution (Supplementary Fig. 12).

The NMR-derived ATP-lid structure in the closed state of HSP90α-NTD was compared to the NTD extracted from full-length HSP90α (FL-HSP90-NTD) focusing on the ATP-lid segment that is stabilized in the closed conformation by the binding of ATP, co-chaperone and client protein[66]. The ATP-lid closed state of apo-HSP90α-NTD can be superimposed on the ATP-lid of dimeric full-length HSP90α (PDB 7L7J) with a RMSD of 4.1 Å (Cα atoms). This RMSD value highlights the

human HSP90α-NTD. The transition pathway between open and closed ATP-lid states, distant by up to 30 Å, is obviously more complicated than a simple two-state model used here for the quantitative analysis of our CPMG relaxation-dispersion data and we cannot exclude that intermediate states could also be involved in the broadening of NMR signals. A more detailed analysis of the computed MD trajectories allows to obtain further information on the kinetics of the

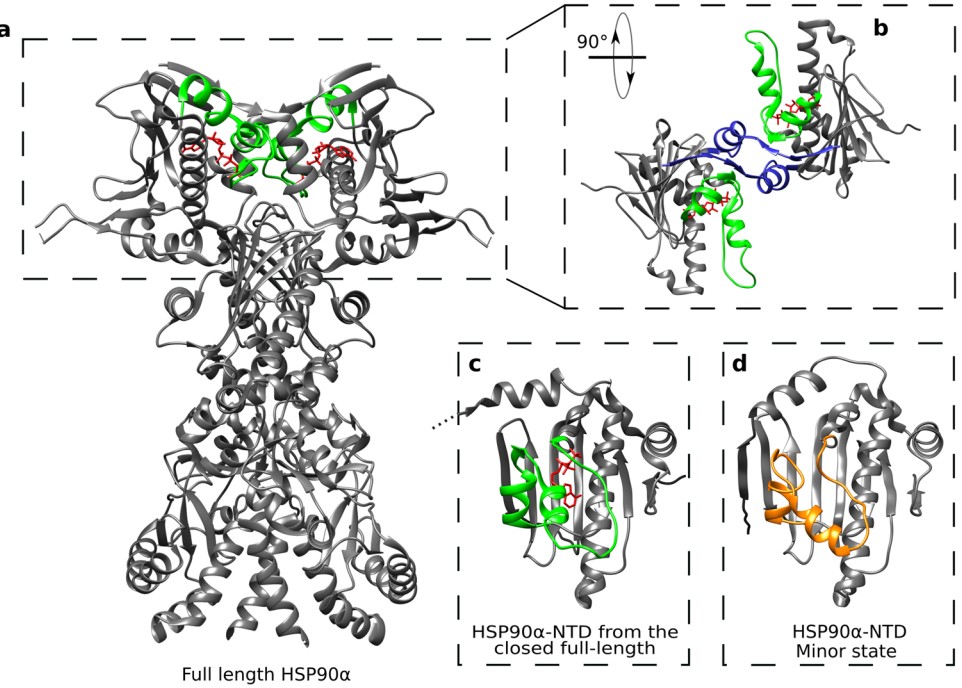

**Fig. 6 | Structure comparison of HSP90α-NTD excited state with full-length dimeric HSP90 α functional cycle intermediate. a** Structure of the homodimer full-length HSP90α in closed form stabilized by p23, FKBP51 and ATP (PDB 7L7J)[66]. The segment covering the nucleotide-binding site and ATP are colored in green and red, respectively. **b** Zoom on the two N-terminal domains of the full-length HSP90α, turned by 90°. The blue segments represent the β strands exchanging between the two chains of the homodimer. **c** N-terminal domain of HSP90α (PDB 7L7J). **d** Average structure of the calculated ensemble for ATP-lid closed (excited) state of apo HSP90α-NTD. The segment 98–136 is represented in orange.

resemblance of the NMR-observed metastable excited state to the structure of the ATP-lid in FL-HSP90-NTD (Fig. 6c–d). In particular, the secondary structural motifs describing the ATP-lid segment are comparable: helix-3 and helix-4 are present, while the segment forming helix-5 in the open conformation is in an extended conformation in both FL-HSP90-NTD and the excited state of isolated HSP90-NTD. The presence of ATP in the structure of the full-length protein as well as the β-strand exchange leading to additional interactions may explain the remaining differences. Although our study was performed on isolated HSP90α-NTD, our results suggest that the functionally relevant ATP-lid closed state of HSP90 ATP-lid pre-exists in apo HSP90 before dimerization driven by ATP binding. Therefore, ATP binding selects and stabilizes a preexisting conformation before inducing further structural rearrangements. Our key finding is the characterization of the apo-HSP90α N-terminal domain conformations sampled in solution, and the corresponding interconversion rates and populations. Moreover, our results support that the closing of the ATP-lid is not an induced fit due to the binding of a nucleotide as the closed ATP-lid conformation is already present in apo protein. These results highlight the importance of investigating protein dynamics and structure at a physiologically relevant temperature. This is particularly relevant for the structure-based design of new drugs against the ATP binding domain of human HSP90.

## Methods
### Preparation of isotopically labeled HSP90α-NTD samples
*E. coli* BL21-DE3-RIL cells transformed with a pET-28 plasmid encoding the N-terminal domain of HSP90α from *Homo Sapiens* (HSP90α-NTD) with a His-Tag and a TEV cleavage site were progressively adapted in three stages over 24 h to M9/$^2$H$_2$O. In the final culture, bacteria were grown at 37 °C in M9 medium with 99.85 % $^2$H$_2$O (Eurisotop), 1 g/L $^{15}$N$^1$H$_4$Cl (Sigma Aldrich) and 2 g/L D-glucose-d$_7$ or D-glucose-$^{13}$C$_6$-d$_7$ for U-[$^2$H, $^{12}$C, $^{15}$N] and U-[$^2$H, $^{13}$C, $^{15}$N] HSP90α-NTD samples,

respectively. For production of U-[$^2$H, $^{15}$N, $^{12}$C], Ala-[$^{13}$C$^1$H$_3$]$^β$, Met-[$^{13}$C$^1$H$_3$]$^ε$, Leu-[$^{13}$C$^1$H$_3$]$^{δ2}$, Ile-[$^{13}$C$^1$H$_3$]$^{δ1}$, Thr-[$^{13}$C$^1$H$_3$]$^γ$, Val-[$^{13}$C$^1$H$_3$]$^{γ2}$ wild type HSP90α-NTD or mutants (R46A, R60A, D127A, S129A), HLAM-A$^β$I$^{δ1}$M$^ε$(LV)$^{proS}$T$^γ$ labeling kits (NMR-Bio) were added to the M9/$^2$H$_2$O media[50]. To cross-validate sequence-specific assignment of methyl probes, 56 single-point mutant samples were produced[51] (see Supplementary Fig. 2 for list of mutants).

For all samples, protein production was induced by 0.5 mM of IPTG when the O.D at 600 nm reached ca. 0.8, overnight at 20 °C. HSP90α-NTD samples were purified in two steps using a Ni-NTA affinity chromatography step followed by a size exclusion chromatography step[49]. For relaxation dispersion experiments, an additional tag cleavage step was performed for the preparation of the wild type HSP90α-NTD samples, by the addition of TEV protease at room temperature for 2 h (in a ratio 1:10) before the removal of His-tagged-TEV protease and cleaved His-Tag using reverse Ni-NTA affinity chromatography. SEC-MALS and SDS-PAGE followed by Coomassie staining, were used to monitor sample homogeneity.

### NMR spectroscopy
The final NMR buffer conditions were 20 mM HEPES (pH 7.5), 150 mM NaCl and 1 mM TCEP in either 100% $^2$H$_2$O or 90% $^1$H$_2$O/10% $^2$H$_2$O. Single point mutant samples to cross-validate the assignment were concentrated at [0.1–0.4] mM. 40 μL of each sample was loaded in a 1.7 mm NMR tube and 2D $^1$H-$^{13}$C SOFAST methyl TROSY[67] experiments were recorded for an average duration of ~1.5 h each, on a spectrometer equipped with a 1.7 mm cryogenically cooled, pulsed-field-gradient triple-resonance probe and operating at a $^1$H frequency of 850 MHz. All other NMR experiments were recorded on isotopically labeled HSP90α-NTD samples (200 μL of 0.5 mM labeled protein in 4 mm Shigemi tube) using a Bruker Avance III HD spectrometers equipped with 5 mm cryogenic probes, Topsin 3.5 and NMRlib 2.0 softwares.

3D HNH- and NNH-edited NOESY experiments were recorded at 293 K on a spectrometer operating at a $^1H$ frequency of 850 MHz using a U-[$^2H$, $^{13}C$, $^{15}N$] wild-type HSP90α-NTD sample in 90% $^1H_2O$/10% $^2H_2O$ buffer. 3D CCH-edited HMQC-NOESY-HMQC NMR experiments[68,69] were recorded on high field spectrometer, operating at a $^1H$ frequency of either 850 MHz or 950 MHz, using U-[$^2H$, $^{15}N$, $^{12}C$], Ala-[$^{13}C^1H_3$]$^β$, Ile-[$^{13}C^1H_3$]$^{δ1}$, Leu-[$^{13}C^1H_3$]$^{δ2}$, Met-[$^{13}C^1H_3$]$^ε$, Thr-[$^{13}C^1H_3$]$^γ$, Val-[$^{13}C^1H_3$]$^{γ2}$ samples of wild type, R46A-, R60A-, or S129A-HSP90α-NTD constructs in $^2H_2O$ buffer. 3D CCH-edited NOESY were also acquired with wild type HSP90α-NTD in the presence of an excess of ADP or resorcinol ligands. Each 3D CCH-edited NOESY experiment was recorded over ca. 2–3 days, at 298 K for wild type HSP90α-NTD, 288 K for R46A- and S129A mutants, or 293 K for R60A-HSP90α-NTD.

Methyl TROSY $^{13}C^1H$-multiple quantum CPMG relaxation dispersion experiments[70] modified with Ernst angle excitation[67] experiments and $^{15}N$ single quantum CPMG relaxation dispersion experiments were recorded at 293 K using U-[$^2H$, $^{15}N$, $^{12}C$], Ala-[$^{13}C^1H_3$]$^β$, Ile-[$^{13}C^1H_3$]$^{δ1}$, Leu-[$^{13}C^1H_3$]$^{δ2}$, Met-[$^{13}C^1H_3$]$^ε$, Thr-[$^{13}C^1H_3$]$^γ$, Val-[$^{13}C^1H_3$]$^{γ2}$ WT- HSP90-NTD sample, on spectrometers operating at $^1H$ frequencies of 700 MHz and 850 MHz ($^{13}C^1H_3$ experiments) or of 600 MHz, 700 MHz and 850 MHz ($^{15}N$ experiments). $^{13}C^1H$-multiple quantum CPMG relaxation dispersion experiments were also acquired with HSP90α-NTD-WT, R46A- and R60A-HSP90α-NTD samples at 288 K. Methanol was used as a reference for precise temperature calibration on each spectrometer. The power of the $^{13}C$ refocusing pulse in CPMG was set to 16 kHz (or 4 kHz for $^{15}N$), the CPMG relaxation period was set to 30 ms for $^{13}C$- (60 ms for $^{15}N$-) experiments. Dispersion profiles, recorded in an interleave manner, comprised 16 different CPMG frequencies ($υ_{cpmg}$) and were ranging from 66 Hz to a maximum of 2000 Hz for methyl (Fig. 5b), whilst dispersion profiles comprised 11 different $υ_{cpmg}$ and were ranging from 33 Hz to a maximum of 1000 Hz for $^{15}N$ experiments (Supplementary Fig. 8B). In addition, a reference spectrum was acquired for both types of experiments by omitting the relaxation period. The total experimental acquisition time for the relaxation dispersion experiment was 1 day for $^{13}C$-experiments and 2 days for $^{15}N$ experiments.

## Structural restraints

The following types of experimental distance restraints were used for structure calculations of HSP90-NTD variable segment from M98 to V136: (i) $H_N$–$H_N$ restraints derived from $^{15}N$-edited NOESY spectra acquired using a NOE mixing time of 240 ms; and (ii) methyl −methyl restraints from $^{13}C$-edited NOESY spectra. A qualitative approach was used in the derivation of distance restraints. In particular, intermethyl distance restraints were fixed between the barycenters of each proton triplets with distance bound of 2–4 Å and 4–6 Å for strong and weak cross-peaks detected in 3D-NOESY acquired with a short mixing time (i.e. 100 ms) or with distance bound of 4–8 Å for cross-peaks observed only in experiment acquired with longer NOE mixing time (ca. 300 to 400 ms). In the case of the mutant HSP90-NTD-R46A, intermethyl distance restraints were fixed between the barycenters of each proton triplets with distance bound of 2–8 Å as no short mixing 3D-NOESY experiment was acquired. $H_N$–$H_N$ distance bounds for observed NOE cross-peaks were fixed to 2–6 Å. For R60-D127 and R46-S129 stabilizing interactions identified using mutagenesis, supplementary distances restraints were fixed between oxygen acceptor atoms and arginine side chain nitrogen/hydrogen donor atoms with distance upper limits of 2/3 Å, respectively. Backbone chemical shifts of WT-HSP90-NTD constructs were used to determine φ and ψ dihedral angles using TALOS + software[53,71,72] and angular restraints with a tolerance of ±10° were applied. For the HSP90-NTD segment from M98 to V136, a total of 54 dihedral angles restraints were used for the calculation of WT- and R60A-HSP90-NTD structures. Due to widely broadened missing peaks, no angular restraints were experimentally derived from residues 105 to 115.

For the structure refinement of R46A-HSP90-NTD excited state, the upper limit of the 5 distance restraints specific to the ATP-lid closed state (Supplementary Table 2) was reduced to 6.5 Å to take into account the lower population of this transiently sampled state, while the 22 structural restraints specific of the ATP-lid in open conformation (Supplementary Table 2) were excluded of distance restraints set. The angular restraints for helix−5 (from I128 to V136) were excluded for the calculation R46A-HSP90-NTD structures, as we had evidences from initial structure calculation that helix−3 and helix−4 are preserved while helix−5 unfolds in the excited state (Supplementary Table 6). The total number and distribution of structural restraints can be found in Supplementary Fig. 2 and Supplementary Table 1.

## Structure calculation

Structural calculations were performed using CYANA 3.98.13 simulated annealing protocol[54]. The invariable segments from residues 11 to 97 and from 137 to 223 (Fig. 1) were treated as a rigid body core using a set of 13434 $C_α$-$C_α$ distance restraints and 477 dihedral angles (φ, ψ and $χ_1$) extracted from X-ray crystal structure of apo HSP90-NTD (PDB 1YES)[73]. The solution structures sampled by the variable segment of HSP90-NTD (Fig. 1) from residue M98 to V136 which harbors the specific loop covering the ATP-binding site was determined using NMR structural information obtained on 3 different samples (WT and R46A-, R60A-mutants). Initial structure calculation of WT- and R46A-HSP90-NTD were performed using a two-state structure calculation protocol enabling each distance restraints to be satisfied in either ATP-lid open- or closed-state or both states simultaneously[74].

Structural data acquired for the R60A-construct were used with a single state calculation protocol to refine the ATP-lid open structural ensemble (Fig. 3a). Conversely, the intermethyl NOE data set acquired for R46A-HSP90-NTD construct was used to refine the ATP-lid closed structural ensemble. For each HSP90-NTD structural state, a CYANA simulated annealing protocol using 25,000 torsion angle dynamic steps[54] was used to generate one thousand conformers starting from random coordinates and using NMR experimental restraints. The twenty conformers characterized by the lower target function values were selected for further refinement using a CNS-restrained molecular dynamics protocol using a full Lenard-Jones potential and explicit water molecules[75]. In brief, the explicit solvent refinement consisted of the five following steps: (i) immersion in a 7.0 Å shell of water molecules and energy minimization; (ii) slow heating from 100 to 500 K in 100 K temperature steps with 200 MD steps per temperature step (time/step 3 fs), with harmonic position restraints on the protein heavy atoms that were slowly phased out during the heating stage; (iii) refinement at 500 K with 2000 MD steps (time step 4 fs); (iv) slow cooling from 500 K to 25 K in 25 K temperature steps with 200 MD steps per temperature step (time step 4 fs); (v) final energy minimization (200 steps).

## Molecular dynamics simulations

The experimentally refined ensembles were used as starting points for MD simulations. Forty separate simulations were run for 1 μs, for the 20 different conformations observed for both the ATP-lid in the closed and open states. First, the systems were setup. The all-atom Amber ff14SB force field[76] was used with explicit water molecules described using TIP3P model[77]. The Gromacs software[78] (single precision, 2020 version) was used to perform preparation, equilibration and production steps. Each protein model was solvated in an octahedral box of about 13,000 water molecules extending at least 13 Å away from the protein. Standard protonation states were assumed. The His189 was protonated on the Nδ1 atom, whereas the other three His residues were protonated on the Nε2 atom, in accordance with the pKa calculations performed by H++ server[79] tested on 20 typical structures with relative permittivity of the solvent of 78, the relative permittivity of the protein of 4 at a salt concentration of 170 mM. This resulted in a net

charge of −11 for the protein. NaCl salt and Na$^+$ counterions were added to obtain an electrically neutral system, with a salt concentration of about 170 mM. Then, for each simulation, steepest-descent energy minimization was used until the maximum force was lower than 100 kJ.mol$^{-1}$.nm$^{-1}$, followed by 100 ps constant-volume equilibration and 100 ps constant-pressure equilibration, both performed with heavy nonwater atoms restrained toward the starting structure with a force constant of 1000 kJ.mol$^{-1}$.nm$^{-2}$. Finally, 1020 ns were performed with constant pressure, constant temperature, without restraints. The first 20 ns were considered as an initial equilibration, and the last 1000 ns were used for the analysis presented in the article. All bonds involving hydrogen atoms were constrained to the equilibrium value using 1 iteration to the 4th order of the Linear Constraint Solver[80], allowing for a time step of 2 ps for the leap-frog integrator. The temperature was kept constant at 300 K using two stochastic velocity rescaling thermostats[81] for the solvent and for the protein, with a time constant of 2 ps. The pressure calculated with long-range correction was kept constant at 1 atm using a Parrinello-Rahman isotropic barostat[82] with a relaxation time of 2 ps and a compressibility of $4.5 \times 10^{-5}$ bar$^{-1}$. Long-range electrostatics were handled by smooth particle-mesh Ewald (PME) summation[83] with a fourth-order B spline interpolation and a grid spacing of 0.16 nm. The cutoff radius for Lennard−Jones interactions was initially set to 10 Å, but the neighbor list parameters were automatically optimized with an energy drift tolerance of 0.005 kJ.mol$^{-1}$.ps$^{-1}$ per particle[84]. The snapshots taken every 1 ns were analyzed with Gromacs and MDAnalysis[85]. Concerning RMSD and clustering, for each of the 40 trajectories, conformations were extracted at a regular spacing in time: every 10 ns, for the pair-wise RMSD plotted on Fig. 4a; every 1 ns, for the clustering procedures illustrated by Fig. 4b, c. Clusters were obtained separately using either the 20 μs simulations starting with the ATP-lid in the open state, or the ones starting with the ATP-lid in the closed state. Clustering analysis was realized using TTClust 4.10.1 to define clusters among the conformations extracted from the 20 μs simulations starting with the ATP lid in either the open or the closed states. First, structures were superimposed on the more rigid backbone, excluding the ATP-lid and the flexible tails of the chain (residues 40–97 and 137–220). The pair-wise RMSD matrix was then calculated for the backbone of the ATP-lid (residues 98 to 136), and used to calculate a hierarchical linkage matrix (ward variance minimization algorithm), where the cluster number was optimized automatically[86]. Concerning the violations, either for the closed or for the open state, the specific NOEs are defined by an ensemble of N couples of residues {ij} (supplementary Table 2). At each time step, the distances $d_{ij}$ between the methyles are measured, and the violation V relative to a given state is the average over the N violations $v_{\{ij\}}$, i.e. $V = 1/N \sum_{\{ij\}} v_{\{ij\}}$, where $v_{ij} = \max(0, d_{ij} - d_{ij}^{viol})$. The distance $d_{ij}$ between two protonated methyles is the distance between the two centers of masses of the three respective protons. We have chosen to use $d_{ij}^{viol} = <d_{ij}> + 2\,\sigma(d_{ij})$, where $<d_{ij}>$ and $\sigma(d_{ij})$ are the average value and the standard deviation from the bundle of the 20 structures obtained after refinement under NMR restraints, respectively. Supplementary Tables 2a, b provides the details on the couples {ij}, the three protons defining the center of masses, and their respective $d_{ij}^{viol}$. Concerning the secondary structure propensities along the sequence, calculation were done using DSSP on 120 frames per simulation, extracted from the last 120 ns of the simulation[87]. The three helical types (α, 3$_{10}$ and π) were merged into a single type called "helices".

### Relaxation dispersion data analysis
Peak intensities for relaxation-dispersion CPMG experiments were determined using the nonlinear fitting routine nlinLS from the NMRPipe software suite[88]. Relaxation dispersion profiles were generated from peak intensities, measured from pseudo-3D experiments using different CPMG frequencies and divided by intensities extracted from the reference spectrum[70]. After visual inspection to exclude too noisy relaxation dispersion curves, relaxation dispersion data of the non-overlapping peaks for which exchange contribution (R$_{ex}$) to the transverse relaxation was ≥2 s$^{-1}$ were analyzed and fitted with the software ChemEx[89] using a two-state exchange model. Errors for R$_{2,eff}$ rate values were estimated from twice the noise measured in the spectra. However, when errors were less than 2% of the R$_{2,eff}$ rate value, an error of 2% was assumed[70].

First, a grid search using relaxation data was carried out where both conformational exchange rate and population of the excited state were fixed (Supplementary Fig. 8a) to identify the minimum in order to select these parameters as starting parameters for the global numerical fitting (a single and identical local minimum was identified for both $^{15}$N and $^{13}$CH$_3$ relaxation datasets). For each experimental temperature, a global numerical fit of $^{13}$CH$_3$-CPMG relaxation dispersion profiles was performed to obtain global exchange parameters (exchange rate (k$_{ex}$) and the population of the excited state (P$^*$)) as well as residue-specific parameters (absolute chemical shift difference between the ground and the excited state of the system under chemical exchange, |Δω$_C$| and |Δω$_H$|). A jackknife resampling was run to ensure that all residues globally fitted were part of the same exchange process, and finally 40 Monte-Carlo repeats were used to estimate precision on global fit results. Experimental data for 21 different methyl probes were used for the global fitting of the methyl $^{13}$C-$^1$H CPMG-RD experiments acquired at 293 K.

### Reporting summary
Further information on research design is available in the Nature Portfolio Reporting Summary linked to this article.

### Data availability
All the experimental NMR data used for the structural and dynamics characterization of human HSP90-NTD have been deposited in BMRB under the code bmrbig44. The structures ensembles together with corresponding structural restraints used for the calculation of HSP90-NTD in ATP-Lid open (R60A mutant) and closed (R46A mutant) have been deposited in the PDB with the accession codes 8B7I and 8B7J. The following PDB entries were also used during the course of this study: 7L7J, 1YES, 2XHT, 2XK2, 4NH8, 3K99, 3R4P, 4YKW, 6B9A, 1BYQ. The molecular dynamics trajectories and analysis have been deposited on Zenodo, https://doi.org/10.5281/zenodo.6606744. Source data are provided with this paper.

### Code availability
The various scripts used for the analysis of MD trajectories have also been deposited on Zenodo, https://doi.org/10.5281/zenodo.6606744.

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

## Acknowledgements

The authors thank Drs. R. Awad, C. Laguri, C. Mas and A. Vermot for help and advices. This work used the high field NMR, biophysics characterization, and isotopic labeling facilities at the Grenoble Instruct-ERIC Center (ISBG; UAR 3518 CNRS-CEA-UGA-EMBL) within the Grenoble Partnership for Structural Biology (PSB). Platform access was supported by FRISBI (ANR-10-INBS-05-02) and GRAL, a project of the University Grenoble Alpes graduate school (Ecoles Universitaires de Recherche)

CBH-EUR-GS (ANR-17-EURE-0003). IBS acknowledges integration into the Interdisciplinary Research Institute of Grenoble (IRIG, CEA). Authors acknowledge support from the PSMN (Pôle Scientifique de Modélisation Numérique) of the ENS de Lyon and from GENCI/TGCC (grant A0110807662) for the computing resources. This work was supported by grants from CEA/NMR-Bio (research program C24990), by the Region Auvergne-Rhône-Alpes (pack ambition recherche 2019 - P089), by the French National Research Agency in the framework of the "Investisse-ments d'avenir" program (ANR-15-IDEX-02) and by a fellowship (to F.H.) from "La Ligue contre le Cancer".

## Author contributions

C.L., E.R., F.H., J.B., and M.F. designed the experiments; E.C. and F.H. prepared the samples; F.H., J.B., B.B., and P.M. set up and collected the NMR experiments; E.R., F.H., J.B., and P.G. analyzed the NOESY experi-ments; A.F. and E.R. performed structural calculations; F.H., J.B., and P.M. analyzed the CPMG experiments; C.L., E.R., and P.J. performed and analyzed the MD simulations; C.L., E.R., F.H., and J.B. wrote the manu-script. All authors discussed the results, corrected the manuscripts, and approved the final version.

## Competing interests

The authors declare no competing interests.
