## [Peer Review File · Nature Communications]

Visualizing the Transiently Populated Closed-State of Human HSP90 ATP Binding DomainReviewers' Comments:

Reviewer #1:

Remarks to the Author:

The authors characterize the ATP-binding domain, the N-domain of HSP90. HSP90 is an important molecular chaperone implied in many forms of cancer. The N-domain studied here encompasses the client and cochaperone binding site. ATP binding and subsequent hydrolysis are functionally important to induce chaperon function. Developing ATP-competitive inhibitors has remained a validated drug target. Such development has been hampered in the past by the complex molecular architecture of HSP90 harboring a flexible ATP-lid segment that can occlude the ATP binding site.

The authors, for the first time, characterize besides an open conformation a second, minor closed conformation of the apo state of the N-terminal domain of HSP90. Transition between the conformations requires 30 Angstrom large motions. The massive conformational change can be best seen in Figure 3a and 3b, showing the complete "turn-around" of major helical segments within the NTD-domain. Thus, the functionally important closed conformation of the chaperone is already preformed in the apo state of the molecular chaperone. Such minor state may be addressable by drug inhibitors, the findings reported here are thus highly relevant for drug design.

A highlight of the manuscript is the derivation of the thermodynamics and kinetics describing asymmetric two-site exchange between the two NTD-domain conformations (Figure 5,c).

To be able to extensively characterize the NTD-domain, the authors produced perdeuterated samples with side-methyl protonated sites together with extensive mutagenesis. This allowed NMR resonance assignment of previously unassigned residues (references 36-38) and the detection of extremely long-range NOEs of more than 10 Angstrom distance only detectable in the samples of such labeling scheme.

The existence of the two conformations as well as the key interaction that drive the populations are convincingly investigated and supported by mutagenesis.

The manuscript is well written, the findings are very timely. The quality of the conducted experiments and their interpretation are technically sound. I thus recommend publication of the manuscript after a single minor point has been addressed as outlined below.

Minor revisions:

Figure Caption Figure 5: Replace * with dagger symbol.

Reviewer #2:

Remarks to the Author:

In this article Henot, Rioual et al. use NMR spectroscopy to show for the first time that the NTD of human Hsp90 adopts at least two conformations in solution and in absence of nucleotide. The authors use specific labeling of methyls that were assigned previously (Henot et al. 2021) to unambiguously show that the observed NOE network can only be explained by two conformations of the lid region: a known open conformation and a closed conformation similar to the one observed previously in HSP90-FL with nucleotides. The authors then determine the solution structures of these two conformations, and further validate them using specific mutations. They finally characterize their populations and their exchange rate using methyl-CPMG experiments and MD simulations. While the closed-conformation of the lid is not new, it is the first time that it is observed in the isolated NTD and more importantly, without nucleotides.

Overall, the article is well written and shows novel results using technically sound approaches. It highlights particularly well the power of NMR and methyl specific labeling to identify and characterize low populated states of biological relevance, even the case of an extensively studied protein for which hundreds of solved structures only show one main conformation. In my view, the main point that could be improved is the relation between the current results and the effect on ligand binding. From

full-length Hsp90, we know that lid closing is related to nucleotide binding. Therefore, it would have been very interesting to also investigate the effect of this minor conformation of the NTD on nucleotides or inhibitors interactions with Hsp90 and inversely the effect of ligand binding on the conformation of the lid.

I only have a few specific comments and questions:

Major comments :

- I believe the manuscript will be strongly reinforced by investigating the influence of one or some nucleotides and/or common inhibitors on the closing of the lid. Repeating the CPMG experiments with bound-NTD would probably be unrealistic, but maybe the NOESY spectra or simply the position of the peaks in the HMQC of the residues affected by exchange and outside the binding pocket could provide an answer? As it stands, we can't really conclude on the impact of this minor conformation on ligand binding, which is the main problematic laid in the introduction of this article. Does it favor locking ligands in the domain or instead prevent their entry in the pocket. Maybe simply looking at ligand binding affinities with NTD WT, R60A and R46A could yield interesting results. While these experiments are not required for the current story, I think they would greatly improve the impact of the work.
- Of course, It would also be interesting to have some hints on whether the minor population is also present in larger constructs of Hsp90, full-length or simply including the middle domain. But this would require significant amount of work and is not expected for the current work.

Minor comments :

- In the first paragraph, the authors compare the known structures of Hsp90a NTD. Visually, I cannot see a single difference between the groups (1) (2) and (8). This is also confirmed by their RMSD close to 0. So why are these separated and not regrouped into one single class?
- Is it possible to assign these structure clusters to certain ligand bound? Is it that all apo structures are in the same groups and all nucleotides too? Would it make more sense to only look at the variability of the apo structures to have a better idea of the flexibility of the NTD lid without ligands?
- In the end, which cluster is validated for the solved major conformation? That would be interesting to know, since the authors already did the work. It looks to me like groups 1,2,8 but I don't think it's discussed at all?
- Is the known structure with closed lid from FL Hsp90 consistent with the NOE network observed by the authors? Could it be the actual structure but with too much ambiguity from the NOEs or there are actual violations? That could indicate whether the nucleotide has a direct effect on the lid conformation.
- In the discussion: I don't completely understand the point of the evolutionary coupling analysis? The existence of the close-state lid in FL Hsp90 has been well known for years and documented by multiple cryo-EM structures, so we don't really learn much from the EC?

Reviewer #3:

Remarks to the Author:

In this study titled "Visualizing the Transiently Populated Closed-State of Human HSP90 ATP Binding Domain", Henot et al. determine the structure of the open and closed conformations of Hsp90 N-terminal domain. The authors further show that both of these structures exist simultaneously in the WT NTD of Hsp90, exchanging at a rate of $\sim 2500 \text{ s}^{-1}$, with the closed state populated to $\sim 3\%$.

This is a very well executed structural study, which exploits cleverly designed mutations that stabilize either the closed or the open states of the protein, allowing the authors to calculate atomic resolution structures for both states.

The structural characterization of the newly identified closed conformation can be of potentially great interest to both the understanding of the fundamental mechanism of function of Hsp90 protein, as well as for designing specific inhibitors targeting the NTD domain.

Overall the NMR data is of very high quality and the manuscript is very well written. I do, however, have two major comments -

1) The authors used CPMG RD to study the exchange between the two states in the wild-type protein. These experiments should also be repeated for the R60A and R46A mutants in order to validate that the mutations indeed influenced the population of the two states. In addition, there is currently no experimental evidence showing that the excited state detected by the CPMG RD is indeed the open state of the protein. As the exchange determined by CPMG RD is quite fast for such large conformational changes, this point must be further validated. The authors can compare the chemical shifts of the excited state to see if these indeed resemble the closed state conformation.

2) This study was performed with an isolated NTD domain, and there are no experiments tying these findings to either the structure or the function of the full length Hsp90 protein. In my opinion, such experiments are crucial to prove the biological importance of the newly found closed state of the NTD. Likewise, the manuscript states that the structure of the closed state precludes ATP binding, however this statement does not appear to be supported by experimental evidence.

Minor comments-

1) All the experiments in this study were performed without a nucleotide. Can the authors speculate how the results would change if ATP or ADP were included in their measurements?

2) In the methods sections it is stated that both methyl-carbon and nitrogen CPMG experiments were recorded, however no data for the nitrogen experiments is shown.

3) In the discussion section it is stated that the R46A mutant of Hsp90 stays monomeric in solution, however no experimental evidence (such as SEC-MALS for example) is provided.

Reviewer #4:

Remarks to the Author:

In the paper " Visualizing the Transiently Populated Closed-State of Human HSP90 ATP Binding Domain " the authors identified a metastable excited state in the isolated HSP90 ATP binding domain using solution NMR and mutagenesis. They demonstrated that in solution the HSP90 ATP binding domain transiently samples a functionally relevant ATP-lid closed state, distant by more than 30 Å from the ground state.

Major Points

1. There is an enormous literature on the subject of this manuscript (both computational and experimental) that is only very briefly mentioned in Introduction. Even given page limits, the authors should have critically assessed the previous studies and, more importantly, identify key issues and questions unanswered thus far.

2. The authors claim that this is the first time that such an excited closed-state structure is reported for isolated apo HSP90-NTD. They suggest that the closure of the ATP-lid, observed during the ATP-driven functional cycle of HSP90, is already sampled in the human chaperone before binding of ATP.

Nanosecond single-molecule fluorescence studies uncovered a two-step mechanism for lid closure over the nucleotide-binding pocket and reported structural changes in the lid that are induced by ATP binding. These studies were conducted for the complete Hsp90 dimer
<https://www.ncbi.nlm.nih.gov/pmc/articles/PMC4955915/>

The relevance of the proposed structural models in the context of the functional Hsp90 dimer is not clear and suggestion that the ATP-lid closed state is a metastable excited state requires a more detailed investigation. The important question is whether observed excited states of the chaperone lid would be preserved on the conformational landscapes of the fully functional Hsp90 dimer. The abundance of structural and dynamic data about Hsp90 conformations may be used to establish a reasonable model of the Hsp90 dimer to match the observed lid variations.

3. Structural and dynamic analyses from MD simulations are incomplete and superfluous, missing steps of analyzing convergence of sampling, correspondence with the crystal structures, global dynamics, etc. None of these questions were raised in the manuscript. In general, computational results are not adequately justified, which weakens their connection with the solid experimental data.

4. The integration of different tools in a cohesive and robust pipeline is a crucial point of this paper. However, the overall flow of the approach is not well presented, and some details remain fairly elusive. The authors should address this point.

5. Could the authors clearly identify what makes their findings novel? What do the results of this study add to our current knowledge of the role of Hsp90 dynamics in catalytic mechanisms? It currently looks like a useful incremental improvement.

6. I believe that the authors should spend some time thinking how to strengthen the interface between experiment and computations in the manuscript in order to substantiate key findings.

7. The predicted excited state could have been validated by potentially identifying small molecules that would shift the equilibrium and stabilize the new conformation. The abundance of Hsp90-NTD inhibitors may be used to screen for compounds compatible with the excited state. The experimental validation of such predictions would not only validate the relevance of the excited state but provide useful therapeutic insights.

7. The manuscript is lacking a systematic statistical framework for assessing significance and quality of predictions. The authors should more clearly formulate and apply their statistical instruments along a common strategy to provide more confidence of quality and reproducibility of their results.

Manuscript NCOMMS- 22-12045

Revised version

Point-by-point response to the reviewers' comments

Colour code:

Black - Reviewers' remarks

Green - authors' comment

red- updated parts of the manuscript

Gray - non-modified parts of the manuscript

Reviewer #1 (Remarks to the Author):

The authors characterize the ATP-binding domain, the N-domain of HSP90. HSP90 is an important molecular chaperone implied in many forms of cancer. The N-domain studied here encompasses the client and cochaperone binding site. ATP binding and subsequent hydrolysis are functionally important to induce chaperon function. Developing ATP-competitive inhibitors has remained a validated drug target. Such development has been hampered in the past by the complex molecular architecture of HSP90 harboring a flexible ATP-lid segment that can occlude the ATP binding site.

The authors, for the first time, characterize besides an open conformation a second, minor closed conformation of the apo state of the N-terminal domain of HSP90. Transition between the conformations requires 30 Angstrom large motions. The massive conformational change can be best seen in Figure 3a and 3b, showing the complete "turn-around" of major helical segments within the NTD-domain. Thus, the functionally important closed conformation of the chaperone is already preformed in the apo state of the molecular chaperone. Such minor state may be addressable by drug inhibitors, the findings reported here are thus highly relevant for drug design.

A highlight of the manuscript is the derivation of the thermodynamics and kinetics describing asymmetric two-site exchange between the two NTD-domain conformations (Figure 5,c). To be able to extensively characterize the NTD-domain, the authors produced perdeuterated samples with side-methyl protonated sites together with extensive mutagenesis. This allowed NMR resonance assignment of previously unassigned residues (references 36-38) and the detection of extremely long-range NOEs of more than 10 Angstrom distance only detectable in the samples of such labeling scheme.

The existence of the two conformations as well as the key interaction that drive the populations are convincingly investigated and supported by mutagenesis.

The manuscript is well written, the findings are very timely. The quality of the conducted experiments and their interpretation are technically sound. I thus recommend publication of the manuscript after a single minor point has been addressed as outlined below.

Minor revisions:

Figure Caption Figure 5: Replace * with dagger symbol.

We thank the reviewer for the comments. In the caption of figure 5 ΔG^* has been replaced by ΔG^\ddagger .

Reviewer #2 (Remarks to the Author):

In this article Henot, Rioual et al. use NMR spectroscopy to show for the first time that the NTD of human Hsp90 adopts at least two conformations in solution and in absence of nucleotide. The authors use specific labeling of methyls that were assigned previously (Henot et al. 2021) to unambiguously show that the observed NOE network can only be explained by two conformations of the lid region: a known open conformation and a closed conformation similar to the one observed previously in HSP90-FL with nucleotides. The authors then determine the solution structures of these two conformations, and further validate them using specific mutations. They finally characterize their populations and their exchange rate using methyl-CPMG experiments and MD simulations. While the closed-conformation of the lid is not new, it is the first time that it is observed in the isolated NTD and more importantly, without nucleotides.

Overall, the article is well written and shows novel results using technically sound approaches. It highlights particularly well the power of NMR and methyl specific labeling to identify and characterize low populated states of biological relevance, even the case of an extensively studied protein for which hundreds of solved structures only show one main conformation. In my view, the main point that could be improve is the relation between the current results and the effect on ligand binding. From full-length Hsp90, we know that lid closing is related to nucleotide binding. Therefore, it would have been very interesting to also investigate the effect of this minor conformation of the NTD on nucleotides or inhibitors interactions with Hsp90 and inversely the effect of ligand binding on the conformation of the lid.

I only have a few specific comments and questions:

Major comments :

- I believe the manuscript will be strongly reinforced by investigating the influence of one or some nucleotides and/or common inhibitors on the closing of the lid. Repeating the CPMG experiments with bound-NTD would probably be unrealistic, but maybe the NOESY spectra or simply the position of the peaks in the HMQC of the residues affected by exchange and outside the binding pocket could provide an answer? As it stands, we can't really conclude on the impact of this minor conformation on ligand binding, which is the main problematic laid in the introduction of this article. Does it favor locking ligands in the domain or instead prevent their entry in the pocket. Maybe simply looking at ligand binding affinities with NTD WT, R60A and R46A could yield interesting results. While these experiments are not required for the current story, I think they would greatly improve the impact of the work.

We thank the reviewer for this suggestion, and we decided to acquire 3D NOESY spectra of HSP90 α -NTD in presence of a nucleotide and a resorcinol derivative in order to reinforce the manuscript. In preliminary experiments, we observed that our construct of HSP90 α -NTD conserves a residual ATPase activity. As HSP90-NTD has low affinity for ATP (mM range for HSP90-NTD – Scheibel et al. *J. Biol. Chem* 1997- doi: 10.1074/jbc.272.30.18608) compared to ADP (μ M range – Halpin and Street. *J. Mol. Biol* 2017 - doi: 10.1016/j.jmb.2017.08.005) a large excess of ATP is required to saturate HSP90 α -NTD. But ADP resulting from residual ATPase activity is competing with ATP binding, increasing significantly the complexity of spectra. Even if we already managed to introduce in NMR samples an ATP regeneration system to maintain ATP concentration during the acquisition of NMR data during ca. 2 h (Mas et al 2018 - doi: 10.1126/sciadv.aau4196), this approach is not

yet efficient enough to enable acquisition of 3D-NOESY experiments over 2 to 3 days. We, therefore, decided to collect 3D NOESY experiments in presence of ADP and a resorcinol derivative having a nM affinity for HSP90 α -NTD (Amaral et al. 2017 - doi: 10.1038/s41467-017-02258-w). In both experiments we detected the NOEs specific of the ATP-lid close state with intensities similar to NOEs observed for apo HSP90 α -NTD and a sentence has been introduced in the Discussion section (page 16, lines 9-13):

“Acquisition of 3D-NOESY spectra using methyl-labeled samples of HSP90 α -NTD in presence of ADP or a resorcinol inhibitor reveals that in both cases the NOEs specific of the ATP-lid closed state are observed with intensities similar to HSP90 α -NTD apo state (Supporting Fig. 10), indicating that this minor conformation is preserved in presence of each one of these two ligands.”

Comparison of 1D traces corresponding to M130 and T65 methyl groups for HSP90 α -NTD apo, ADP-bound and resorcinol-derivative-bound forms are presented on Supporting Fig. 10:

Supporting Figure 10: Intensity variation of closed-state-characteristic NOEs in presence of ligands. 1D traces extracted from 3D CCH HMQC-NOESY-HMQC spectra showing M130 as a diagonal peak and its associated NOE cross peaks for apo-HSP90-NTD, HSP90 in complex with a resorcinol derivative: 5-(2,4-dihydroxy-phenyl)-4-(2-fluoro-phenyl)-2,4-dihydro-[1,2,4]triazol-3-one and HSP90-NTD bound to ADP. Additionally, for HSP90-NTD in complex with ADP a 1D trace showing T65 as a diagonal peak and its associated NOE cross peaks is displayed. NOE cross peaks annotated in orange are characteristic of the ATP-lid in closed state. 3D ^{13}C -edited NOESY spectra were acquired during *ca.* 2-3 days each using high-field NMR spectrometers operating at 950 or 850 MHz and at a temperature of 25°C.

- Of course, It would also be interesting to have some hints on whether the minor population is also present in larger constructs of Hsp90, full-length or simply including the middle domain. But this would require significant amount of work and is not expected for the current work.

Despite the significant amount of work, we tried to compare our results on isolated HSP90 α -NTD with full-length HSP90 α , and we have prepared a methyl specifically labelled sample

of full-length HSP90 α . Unfortunately, the 2D methyl TROSY spectrum of the full-length protein is of poor quality (see spectra below acquired at 298 K). We also collected a 3D-NOESY with this sample but we were unable to extract any NOEs, presumably due to the formation of soluble aggregates. While several publications report high quality NMR data collected on full-length human HSP90 β , we were unable to find in the literature similar NMR data for the human isoform α , presumably due to poor quality of the corresponding spectra hampering extraction of any useful information. In the literature, it was nevertheless reported that HSP90 α “*was much more difficult to express and purify from bacterial cell culture in quantities suitable for NMR spectroscopy*” (Park et al 2011 - doi: 10.1038/nsmb.2045), supporting our findings. We are therefore unable to provide any useful NMR information on the full-length construct of HSP90 α .

Minor comments:

- In the first paragraph, the authors compare the known structures of Hsp90a NTD. Visually, I cannot see a single difference between the groups (1) (2) and (8). This is also confirmed by their RMSD close to 0. So why are these separated and not regrouped into one single class?

We thank the reviewer for this comment. Actually, the clustering was made considering all the HSP90 α -NTD backbone C α atoms, and even if the ATP lid domains (from 98 to 136) for cluster 1, 2 and 8 are similar, the segment 64-75 of HSP90-NTD is different which justifies the 3 clusters. We have modified the order of Figure 1 panels and the figure legend in order to clarify this point:

Figure 1: Analysis of available structures of human HSP90 α -NTD. a) Superimposition of 8 centroids representing the 8 clusters describing the 334 structures of isolated HSP90 α -NTD available in the Protein Data Bank (on January the 5th of 2021). Clustering was performed using the MaxCluster program (<http://www.sbg.bio.ic.ac.uk/maxcluster>) with an Average Linkage type of hierarchical clustering and a threshold value of 1.05. In blue is depicted the

segment [98-136] and in red the nucleotide. **b)** Histogram of the averaged pairwise RMSD between $C\alpha$ backbone atoms of the 8 centroids in Å as a function of the residue number (black). Grey bars represent non-assigned backbone residues⁴⁹. On top of the histogram: secondary structure elements as a function of the residue number (α for helices and β for sheets). **c)** Zooms on the segment [98-136], that shows the highest structural variability, for all the 8 centroids superimposed using the Chimera MatchMaker command. PDB ID and number of structures present in each cluster are disclosed next to each centroid. **d)** Table representing pairwise RMSD in Å between $C\alpha$ backbone atoms belonging to the segment [98-136] of each pair of the 8 centroids superimposed using Chimera MatchMaker command (1: 2XHT, 2: 2XK2, 3: 4NH8, 4: 3K99_A, 5: 3R4P_B, 6: 4YKW_A, 7: 6B9A_A, 8: 1BYQ). Going from blue to red the RMSD values increase. Centroids from clusters (1), (2) and (8) are highly similar (low pairwise RMSD values) on the segment [98-136] but these three centroids differ mainly on another segment of the protein [64-75].

- Is it possible to assign these structure clusters to certain ligand bound? Is it that all apo structures are in the same groups and all nucleotides too? Would it make more sense to only

look at the variability of the apo structures to have a better idea of the flexibility of the NTD lid without ligands?

There are only six apo structures of the isolated HSP90 α -NTD without mutation. To represent the largest ensemble of structures accessible to the ATP-lid we decided to consider all the available structures of the isolated HSP90 α -NTD with and without ligands. The six apo structures belong to cluster 3 (PDB codes:1UYL, 5J2V, 1YER, 3T0H, 6GPW) and cluster 2 (1 YES). This information is now provided in the first paragraph of the Results section (page 6, lines 13-16):

“In the six available apo structures (from which five belong to cluster 3 and one to cluster 2), and all nucleotide- or ligand-bound structures of the isolated HSP90 α -NTD, the ATP-lid is in the so-called open state³³ and does not cover the ATPase site and corresponding drug binding site.”

- In the end, which cluster is validated for the solved major conformation? That would be interesting to know, since the authors already did the work. It looks to me like groups 1,2,8 but I don't think it's discussed at all?

Clusters 1, 2 and 8 are close but the closest one is group 5. This is now indicated at the end of the first paragraph of *NMR refinement of structural models for the ATP-lid open and closed states* section (page 11, lines 16-19):

“Compared to previously solved structures of HSP90 α -NTD (Fig. 1a), the final structure ensemble is closer to cluster 5 (C_{α} rmsd 1.45 Å) and clusters 1, 2 and 8 (C_{α} rmsd 1.65 Å in average), while other clusters superimpose with a backbone rmsd higher than 2.3 Å (Fig. 1.c,d).”

- Is the known structure with closed lid from FL Hsp90 consistent with the NOE network observed by the authors? Could it be the actual structure but with too much ambiguity from the NOEs or there are actual violations? That could indicate whether the nucleotide has a direct effect on the lid conformation.

This is an interesting question, but unfortunately the precision of our NMR data and the resolution of EM structure do not allow us to answer this question. From the 5 NOEs specific of the ATP-lid closed state only one is violated by more than 5 Å in the EM structure (THR-65-HG/ALA 124 HB), and from the EM structure only one inter-methyl distance, shorter than 10 Å and specific to the closed state (ILE-128-HD/LEU-64-HD2), is not observed in our NOESY experiments. However, it is difficult to conclude if these differences are due to local rearrangements of side chains containing methyl groups, or structure uncertainties, or direct effects of nucleotide, co-chaperones or HSP90 middle domain on the ATP-lid conformation. Therefore, we decided to not include any discussion regarding this point in the revised manuscript.

- In the discussion: I don't completely understand the point of the evolutionary coupling analysis? The existence of the close-state lid in FL Hsp90 has been well known for years and documented by multiple cryo-EM structures, so we don't really learn much from the EC?

We agree with the reviewer's comment and we decided to remove the discussion and supporting figure corresponding to the evolutionary couplings analysis in the revised version.

Reviewer #3 (Remarks to the Author):

In the paper “Visualizing the Transiently Populated Closed-State of Human HSP90 ATP Binding Domain” the authors identified a metastable excited state in the isolated HSP90 ATP binding domain using solution NMR and mutagenesis. They demonstrated that in solution the HSP90 ATP binding domain transiently samples a functionally relevant ATP-lid closed state, distant by more than 30 Å from the ground state.

Major Points

1. There is an enormous literature on the subject of this manuscript (both computational and experimental) that is only very briefly mentioned in Introduction. Even given page limits, the authors should have critically assessed the previous studies and, more importantly, identify key issues and questions unanswered thus far.

Indeed, the literature on HSP90 is prolific, the Web of Science database provides about 400 articles per year with HSP90 in the title in the last decade. Following the advice of Reviewer #3, we have reviewed more citations, and selected the ones that we have considered the most significant to underline the question that we have studied: the conformational dynamics of the ATP lid in the apo N-terminal domain. We have specifically spotted the ones that show that the conformational dynamics of the ATP-lid plays a role in the chaperone function of HSP90, via allosteric regulation. Here are the new citations that we propose to add, if the Editor allows that we increase the number of citations:

27. Prodromou, C. *et al.* The ATPase cycle of Hsp90 drives a molecular ‘clamp’ via transient dimerization of the N-terminal domains. *EMBO J.* **19**, 4383–4392 (2000).
28. Pearl, L. H. Review: The HSP90 molecular chaperone - An enigmatic ATPase. *Biopolymers* **105**, 594–607 (2016).
29. Huai, Q. *et al.* Structures of the N-terminal and middle domains of E. coli Hsp90 and conformation changes upon ADP binding. *Structure* **13**, 579–590 (2005).
30. Rashid, S., Lee, B. L., Wajda, B. & Spyropoulos, L. Nucleotide Binding and Active Site Gate Dynamics for the Hsp90 Chaperone ATPase Domain from Benchtop and High Field 19F NMR Spectroscopy. *J. Phys. Chem. B* **124**, 2984–2993 (2020).
35. Krukenberg, K. A., Street, T. O., Lavery, L. A. & Agard, D. A. Conformational dynamics of the molecular chaperone Hsp90. *Q Rev Biophys* **44**, 229–255 (2011).
36. Colombo, G., Morra, G., Meli, M. & Verkhivker, G. Understanding ligand-based modulation of the Hsp90 molecular chaperone dynamics at atomic resolution. **105**, 7976–7981 (2008).
37. Reidy, M. & Masison, D. C. Mutations in the Hsp90 N Domain Identify a Site that Controls Dimer Opening and Expand Human Hsp90 α Function in Yeast. *J. Mol. Biol.* **432**, 4673–4689 (2020).
38. Zhang, H. *et al.* A Dynamic View of ATP-coupled Functioning Cycle of Hsp90 N-terminal Domain. *Sci. Rep.* **5**, (2015).
39. Mickler, M., Hessling, M., Ratzke, C., Buchner, J. & Hugel, T. The large conformational changes of Hsp90 are only weakly coupled to ATP hydrolysis. *Nat. Struct. Mol. Biol.* **16**, 281–286 (2009).
40. Ratzke, C., Berkemeier, F. & Hugel, T. Heat shock protein 90’s mechanochemical cycle is dominated by thermal fluctuations. *PNAS* **109**, 161–166 (2012).
41. Southworth, D. R. & Agard, D. A. Species-Dependent Ensembles of Conserved

Conformational States Define the Hsp90 Chaperone ATPase Cycle. *Mol. Cell* **32**, 631–640 (2008).

We propose to modify one sentence (page 4, line 5), and to add the following paragraph to the introduction (page 4, lines 11 to 24 and page 5 lines 6 to 8):

“...Large changes of the ATP-lid conformation also occur during the functional cycle²⁷. Changes in the lid conformational dynamics are linked to chaperone activity²⁸ and nucleotide binding^{27,29–34}. But there is no consensus yet regarding the driving forces modulating the conformational changes during the chaperone cycle^{28,35}. It was proposed that the lid acts as a nucleotide-sensitive conformational switch of the molecular chaperone activity³⁶. There would be a strong correlation between the chaperone activity and the ATP vs. ADP binding^{30,37,38}. In contrast, it was also reported that the transitions between the conformational states and the nucleotide binding/unbinding are mainly thermally driven, with large conformational fluctuations on timescales faster than the rate of ATP hydrolysis^{39,40}. Last, Southworth et al. mentioned that rather than being irreversibly determined by nucleotide binding, a conformational equilibrium exists between different states⁴¹.

In this article, we report new atomic-resolution structural and **dynamics information on HSP90 α -NTD ATP-lid** in solution. ...”

2. The authors claim that this is the first time that such an excited closed-state structure is reported for isolated apo HSP90-NTD. They suggest that the closure of the ATP-lid, observed during the ATP-driven functional cycle of HSP90, is already sampled in the human chaperone before binding of ATP.

Nanosecond single-molecule fluorescence studies uncovered a two-step mechanism for lid closure over the nucleotide-binding pocket and reported structural changes in the lid that are induced by ATP binding. These studies were conducted for the complete Hsp90 dimer <https://www.ncbi.nlm.nih.gov/pmc/articles/PMC4955915/>

The relevance of the proposed structural models in the context of the functional Hsp90 dimer is not clear and suggestion that the ATP-lid closed state is a metastable excited state requires a more detailed investigation. The important question is whether observed excited states of the chaperone lid would be preserved on the conformational landscapes of the fully functional Hsp90 dimer. The abundance of structural and dynamic data about Hsp90 conformations may be used to establish a reasonable model of the Hsp90 dimer to match the observed lid variations.

Previous results by Schultze et al., *Nat. Chem Biol.* 2016 or PMC 4955915) were already cited (reference number 26) in the second paragraph (page 16 line 21 to page 17 line 2) of the discussion in our originally submitted version:

“PET fluorescence quenching experiments by Schulze *et al.*²⁶ monitoring yeast HSP90-NTD ATP-lid structural rearrangement in isolated monomeric constructs reveals a motion of the ATP-lid with a characteristic rate of 1.5 kHz, even though a full closure of the lid over the nucleotide/drug binding site has not been observed”.

We don't want to push further the comparison as the yeast and human HSP90 α sequences are too different for quantitative comparison and full-length dimeric human HSP90 α does not give rise to useful NMR spectra hampering quantitative analysis (see

answer to reviewer 2). Nevertheless, to avoid confusion we have removed from the introduction our claim “~~that this is the first time that such an excited closed-state structure is reported~~” (page 5, lines 7-8) and we have now indicated at many locations in the manuscript that our results and discussion apply to human HSP90 α -NTD.

3. Structural and dynamic analyses from MD simulations are incomplete and superfluous, missing steps of analyzing convergence of sampling, correspondence with the crystal structures, global dynamics, etc. None of these questions were raised in the manuscript. In general, computational results are not adequately justified, which weakens their connection with the solid experimental data.

In the article, the NMR experiments bring information about the presence of two conformers, but it is difficult, based on these experimental data, to determine which of the two structures is the ground state. The MD simulations were used to investigate the relative stability of the two experimentally-obtained structures (closed and open ATP-lid). In particular, we raise the question whether the two refined models obtained using NMR are stable during MD simulations in absence of experimental restraints. To avoid confusion, we have indicated the information extracted from the MD in the abstract (page 2, lines 10 to 12) and in the introduction (page 5, lines 6 à 8) by changing two sentences:

Abstract modification: “NMR relaxation enabled to derive information on the kinetics and thermodynamics of this interconversion, while molecular dynamics established that the ATP-lid in closed conformation is a metastable excited state”

Introduction modification: “Using solution NMR spectroscopy ~~and molecular dynamics (MD) simulation~~, we could determine atomic-resolution models of the open- and closed-ATP-lid state conformations, as well as derive kinetic and thermodynamic information for this structural rearrangement occurring in solution. ~~Complementary molecular dynamics investigation revealed that the conformation of the closed state is metastable on the microsecond time scale.~~”

Regarding the analysis of convergence, it is often done through the RMSD, for example using the RMSD as a function of time. We provide this information through the RMSD matrix, that is reported in Fig. 4a. In the main text (page 12, line 20 to page 13 line 4) we discuss the pair-averaged C α RMSD of the lid over the 20 000 ns of simulations, either for the closed-lid state (9.3 Å) or for the open-lid state (4.0 Å). The higher value for the closed-state reveals lower stability of the closed-lid state, that shows transition towards different states. However, it is not really possible to prove convergence in the present case. Despite that the length of our simulations (40 x 1000 ns, with 20 describing the open-lid state, and 20 describing the closed-lid state) is very long compared to simulations found in older publications (typically hundreds of ns), a 1 μ s simulation remains too short to describe the full closing/opening equilibrium of the ATP-lid, that takes place in the millisecond timescale. Therefore, our simulations in absence of experimental restraints are not expected to be ergodic and to sample the two states with their respective population. Our aim is not to prove a “convergence” of the sampling. On the opposite, we try to explicitly and honestly describe the complexity of the results, *i.e.* the non-convergence of sampling observed for the closed-lid state in Figure 4. Nevertheless, the stability of the closed-lid state during several 1- μ s-simulations without experimental restraints is a strong support for the existence of an excited metastable state with a closed ATP-lid conformation. To mention this point, we have added one sentence in the Results section (page 13, lines 12-13):

“Analyzing the dynamics of the ATP-lid in individual trajectories reveals three different behaviors that demonstrate that the simulations are not ergodic.”

To improve the structural analysis, we have added a cluster analysis of the trajectories, starting either from the closed state or from the open state, respectively. We have inserted the new analysis in the Fig. 4, see new panels below. The new panels b, c and d report on both the RMSD within the clusters, and the RMSD among the clusters. These RMSDs are larger in the simulations starting from the closed state. **We have added the information on the clustering analysis in the method section (page 26, line 9-19):**

“1.2.1 RMSD and clustering. For each of the 40 trajectories, conformations were extracted at regular spacing in time: every 10 ns, for the pair-wise RMSD plotted on Fig 4a; every 1 ns, for the clustering procedures illustrated by Figs. 4b and 4c. Clusters were obtained separately using either the 20 μ s simulations starting with the ATP-lid in the open state, or the ones starting with the ATP-lid in the closed state. Clustering analysis was realized using TtClust 4.10.1 to define clusters among the conformations extracted from the 20 μ s simulations starting with the ATP lid in either the open or the closed states. First, structures were superimposed on the more rigid backbone, excluding the ATP-lid and the flexible tails of the chain (residues 40-97 and 137-220). The pair-wise RMSD matrix was then calculated for the backbone of the ATP-lid (residues 98 to 136), and used to calculate a hierarchical linkage matrix (ward variance minimization algorithm), where the cluster number was optimized automatically⁸⁶.

We have slightly modified the description of Figure 4 to present the new panels (page 12, line 19 to page 13 line 10):

**Figure 4: Molecular Dynamics investigation of HSP90α-NTD ATP-lid open and closed states. a)** Pairwise RMSD (in nm) among the ATP-lid structures obtained in the forty Molecular Dynamics simulations of 1 µs duration each. The initial models are the ensemble of experimentally refined structures (Fig. 3). Hundred frames, separated by 10 ns from each other, are extracted from each simulation. The first 2000 frames are extracted from simulations starting with ATP-lid open-state conformers (blue square). The last ones are extracted from simulations starting with ATP-lid closed-state conformers (orange square). Going from blue to yellow, the RMSD values increase. **b)** Superimposition of 4 centroids representing the 4 clusters describing all the 20,020 taken conformations during the simulation starting with ATP-lid open-state conformers. This clustering was obtained by TTclust. The ATP-lid centroids are represented in b.1, b.2, b.3 and b.4 with the total number of members for each cluster. Above, a circle represents the spreading of the cluster (in Å). **c)** Same than b, Superimposition of 5 centroids representing the 5 clusters describing all the 20,020 taken conformations during the simulation starting with ATP-lid open-state conformers. The ATP-lid centroids are represented in c.1, c.2, c.3 c.4 and c.5 with the total number of members for each cluster. Above, a circle represents the spreading of the cluster (in Å). **d)** Representation of $C_{\alpha}$ RMSD between the 5 centroids of the 5 clusters obtained for simulation starting with ATP-lid open conformation (orange numbers correspond to those from c). The same is represented for the 4 centroids of the 4 clusters obtained for simulation starting with ATP-lid closed conformation (blue numbers correspond to those from b). Finally, $C_{\alpha}$ RMSD between all the centroids are represented. The red, used to represent $C_{\alpha}$ RMSD, goes darker when the RMSD values increase. **e)** Representation of NOE distance restraints characteristics of ATP-lid open and closed states on the corresponding structure. **f)** Violations of characteristics NOEs monitored during the molecular dynamics simulations performed without restraints, for the 20 simulations starting with the ATP-lid in the open state (See methods for definition of computed violations). The values are averaged over all the ATP-lid open (blue) or closed (orange) states specific distances restraints (Supporting Table 2). **g)** As (c), for a typical simulation starting with the ATP-lid in the closed state that remains stable (I), undergoes a transition towards the ATP-lid in the open-state (II), or derives towards region of conformational space that is neither the closed, nor the open ATP-lid state (III).

Concerning the correspondence with crystal structures, indeed in the initial version of the manuscript, we had not discussed it. We thank the reviewer for this interesting suggestion. In the revised version we have included the RMSD of the cluster centroids obtained from MD simulations relative to the 8 centroids of the clusters described in Figure 1, that are representative of the 344 structures extracted from the PDB, and six HSP90α-NTD apo structures. These new data are presented in Supporting Table 4:

Supporting Table 4: Pairwise RMSD comparison of MD and X-ray representative structures. RMSD between 9 centroids representative of MD trajectories (4 in blue, 5 in orange obtained respectively for the ATP-lid open and closed states as presented on Fig. 4), the 8 centroids representing the PDB structures (Fig. 1) and the 6 available experimental apo structures (PDB codes: 5J2V, 6GPW, 1UYL, 1YER, 1YES, 3T0H) of WT-HSP90-NT. Structures were superimposed on the backbone of residues 11 to 97 and 137 to 220, and RMSD were calculated for C α atoms from residues 98 to 136. The red, used to represent C α RMSD, goes darker when the RMSD values increase. The 4 centroids emerging from the simulations starting from the open lid state have RMSD values between 1.9 and 5.5 Å with the experimental crystal structures. The closest pairs are between the centroid 2 and PDB structures: 2XHT and 2XK2. The 5 centroids representing the simulations starting from the closed lid state can be superimposed on backbone with RMSD values between 3.1 and 18.7 Å. The centroids 1, 2 and 3 represent conformation around the closed state and are very far from the crystal structures. The centroids 4 and 5 represent the conformation that have returned towards the ground state, and are closer to the HSP90 α -NTD representative PDB structures.

In the revised version, we are referring to this supporting Table 4 in the main text (page 13, line 1).

Concerning the reviewer's comment on the global dynamics:

- Focusing on the lid in the main text, we have emphasized the qualitative differences in the dynamics of the open and closed states in Fig 4 (new panels f and g). In particular, the violations as a function of simulation time show some transitions from the closed state to the open state.
- Additionally, to describe what happens also in the rest of the sequence, we have calculated the root mean square fluctuation (RMSF) of the C α positions relative to the average conformation for the simulations that describe either the closed-lid state (4 stable simulations), or the open lid-state (20 stable simulations), as shown on the figure below. The low precision for the closed state is due to the small amount of stable simulations available for this metastable state. Analysis of RMSF values, indicates that the dynamics of the protein is similar for the two states, except for the residues belonging to the ATP-lid. Interestingly, the RMSF profile obtained for the open state is also similar to the one obtained from the PDB structures depicted in Fig. 1d. This figure has been added in the Supporting Fig. 6 and mentioned in the main text with an additional sentence (page 13, lines 10-13).

Supporting Figure 6: Root mean square fluctuation analysis of MD trajectories. RMSF values are first calculated residue-wise for each of the 1 microsecond simulations that remain stable. Then, the average and standard error are calculated for each residue, grouping on the one hand the 20 stable simulations starting from the open lid-state (blue), on the other hand the 4 stable ones starting from the closed lid-state (yellow). For each residue number, the filling is done around the average value with a thickness of two times the standard errors of the mean.

Added sentence to describe the RMSF in main text: (page 13, lines 10-13): “According to the Root-Mean-Square-Fluctuations (RMSF) profiles (see Supporting Fig. 6), the increased conformational fluctuations of the ATP-lid in the closed state does not seem to modify the dynamics of the rest of the protein. Analyzing the dynamics of the ATP-lid in individual trajectories reveals three different behaviors that demonstrate that the simulations are not ergodic.”

Finally, regarding the justification of the computational results, we have summarized in the table below the main results and the corresponding justifications. Here are the correspondences between the results, and the MD data that support them.

Result from MD

Justification

- | | |
|--|--|
|  1. The open-lid state is stable in the MD simulations (20 μs) |  • RMSD matrix of Fig. 4a • average RMSD over the 20 μs cited in main text. • New analysis: Structural Clustering and analysis in Fig. 4 (new panels b, c, d) |
|  2. The closed-lid state is metastable in the MD simulations (20 μs) |  • RMSD matrix of Fig. 4a • Average RMSD over the 20 μs cited in main text. • Variability of the simulation results in Fig. 4g • New analysis: Structural Clustering and analysis in Fig. 4 (new panels b, c, d) |

Result from MD

3. Transitions from the closed to the open state are observed in MD, whereas the opposite transition is not.

4. The closed state is the minor one.

5. The transition from closed to open lid states include several conformational changes, with helix 5 that is slow to reconstruct

6. Global analysis of position fluctuations: the regions of the protein that are not included in the ATP-lid have similar dynamics in the open and closed ATP-lid positions.

Justification

- RMSD matrix of Fig. 4a
- Violations of NOE as a function of time for different simulations: Different tendencies of the violations in Fig. 4f vs. Fig. 4g. Complete data are given in Supporting Information

Deduced from points 1,2,3.

Helicity as a function of amino acid along the sequence at the end of the simulation (supporting Fig. 11)

- **New analysis:** RMSF analysis along the protein sequence. Added as supporting Fig. 6

To conclude, the results of the MD simulations were based on analyses such as pairwise RMSD, and distances as a function of simulation time. Following the reviewer's comments, new analyses have been performed, such as cluster analysis and RMSF within MD trajectories. These were added in the main text or in Supporting Information.

4. The integration of different tools in a cohesive and robust pipeline is a crucial point of this paper. However, the overall flow of the approach is not well presented, and some details remain fairly elusive. The authors should address this point.

We feel that satisfying Reviewer 3 on this point would demand to modify the structure of the manuscript to a large extent. Since Reviewers 1, 2, and 4 seem to be satisfied with the chosen logical flow, we prefer to keep the same structure, but we have included new analyses as demanded by Reviewer 3 (see point 3).

5. Could the authors clearly identify what makes their findings novel? What do the results of this study add to our current knowledge of the role of Hsp90 dynamics in catalytic mechanisms? It currently looks like a useful incremental improvement.

We were surprised by this comment and it is obviously not shared by the three other reviewers. Nevertheless, we have added the following sentence at the end of the discussion section to summarize our main results (page 19, lines 2-6) :

“Our key finding is the characterization of the apo-HSP90 α N-Terminal domain conformations sampled in solution, and the corresponding interconversion rates and

populations. Moreover, our results support that the closing of the ATP-lid is not an induced fit due to binding of a nucleotide as the closed ATP-lid conformation is already present in apo protein.“

6. I believe that the authors should spend some time thinking how to strengthen the interface between experiment and computations in the manuscript in order to substantiate key findings.

In this work, the MD is used specifically to investigate the metastability and relative stability of the two lid positions. Following the suggestions by Reviewer 3, the RMSD calculated between the clusters obtained by MD and experimental structures available in the PDB have been added in Supporting Table 4 and mentioned in main text (see point 3).

7. The predicted excited state could have been validated by potentially identifying small molecules that would shift the equilibrium and stabilize the new conformation. The abundance of Hsp90-NTD inhibitors may be used to screen for compounds compatible with the excited state. The experimental validation of such predictions would not only validate the relevance of the excited state but provide useful therapeutic insights.

In the article, we have studied the impact of relevant mutations on the stability of the two conformations. Moreover, in response to some remarks of Reviewer 2, we have added new NMR data on the HSP90 α -NTD in interaction with ADP and with a resorcinol derivative that show no strong impact of the ligand on the conformational exchange. These new data have been included in the revised article (see answer to reviewer 2)

Unfortunately, investigating systematically the effects of numerous ligands on the conformational exchange is a considerable amount of work (up to years), beyond the scope of the present revision.

7. The manuscript is lacking a systematic statistical framework for assessing significance and quality of predictions. The authors should more clearly formulate and apply their statistical instruments along a common strategy to provide more confidence of quality and reproducibility of their results.

Here are the efforts we have made to improve the clarity concerning the statistical significance and quality of predictions:

- To determine in a statistical manner the minimal number of protein states required to satisfy all the experimental NMR restraints, we used structural correlations value, implemented in PDBcor server, that quantifies the clustering of protein states (Ashkinadze et al. J Biomol NMR 2022 – doi: [10.1007/s10858-022-00392-2](https://doi.org/10.1007/s10858-022-00392-2)). This unbiased statistical analysis confirms that only two states are required to satisfy all the NMR structural information acquired on HSP90 α -NTD in solution. This is now indicated in Results section (page 9, lines 4-11):

“To assess in an automated unbiased manner how many protein states are required to satisfy all the experimental distance restraints, we used structural correlation measure that determines the optimal number of states for multi-state structure calculation that can give more clear-cut results than the conventional target-function-based analysis⁵⁶. The structural correlations value is expected to be maximum when the number of states used for the calculation reaches the number of states sampled by the target protein. The computed structural cor-

relations value is 0.69 for a two states model, and drops to 0.45 for a 3-states model, indicating that a model including two states is sufficient to satisfy all experimental data.”

- The MD simulation data and their analysis have been deposited on a public Zenodo repository. The RMSD matrix and the results of the clustering analysis are available. <https://doi.org/10.5281/zenodo.6606744>

- The statistical methodology is detailed in a reporting form that is deposited with article revised version

- For the RMSF analysis of Supporting Fig. 6, we have indicated in the legend how the error bars are calculated (twice the standard error of the mean obtained on the N simulations, N=4 for the closed state and N=20 for the open state).

Reviewer #4 (Remarks to the Author):

In this study titled “Visualizing the Transiently Populated Closed-State of Human HSP90 ATP Binding Domain”, Henot et al. determine the structure of the open and closed conformations of Hsp90 N-terminal domain. The authors further show that both of these structures exist simultaneously in the WT NTD of Hsp90, exchanging at a rate of $\sim 2500 \text{ s}^{-1}$, with the closed state populated to $\sim 3\%$.

This is a very well executed structural study, which exploits cleverly designed mutations that stabilize either the closed or the open states of the protein, allowing the authors to calculate atomic resolution structures for both states.

The structural characterization of the newly identified closed conformation can be of potentially great interest to both the understanding of the fundamental mechanism of function of Hsp90 protein, as well as for designing specific inhibitors targeting the NTD domain. Overall the NMR data is of very high quality and the manuscript is very well written. I do, however, have two major comments –

1) The authors used CPMG RD to study the exchange between the two states in the wild-type protein. These experiments should also be repeated for the R60A and R46A mutants in order to validate that the mutations indeed influenced the population of the two states.

As requested by the reviewer, we have acquired relaxation dispersion experiments for both R46A and R60A HSP90 α -NTD mutants and observed an increase in the exchange rates k_{ex} to 3800 s^{-1} for R60A and 4200 s^{-1} for R46A. However, it is well established that for fast exchange processes, it is not possible to extract reliable populations from relaxation dispersion data (Kleckner et al. *Biochim. Biophys. Acta* 2011 - doi: 10.1016/j.bbapap.2010.10.012 ; Vallurupalli et al. *J Phys Chem B* 2011 –doi: 10.1021/jp209610v). Nevertheless, if we assume that the R60A mutation (or R46A) will destabilize specifically the closed excited state (respectively open ground state), we expect an increase of k_{-1} (respectively k_1) and therefore an increase of k_{ex} for both mutants as experimentally observed. These new data are now included in Supporting Fig. 9 and introduced at the end of the Results section (page 15, lines 13-20):

“A similar strategy was used to assess how R46A- and R60A- mutations modify kinetics of ATP-lid transition using available $^{13}\text{CH}_3$ -labeled samples (Supporting Fig. 9). Global fits of relaxation-dispersion data collected at 288 K revealed an increase in exchange kinetics for R60A ($k_{\text{ex}}=3764\pm 132 \text{ s}^{-1}$) and R46A ($k_{\text{ex}}=4192\pm 175 \text{ s}^{-1}$) compared to WT HSP90 α -NTD ($k_{\text{ex}}=2994\pm 83 \text{ s}^{-1}$). However, such fast exchange regimes preclude reliable extraction of state populations^{59,60}. Nevertheless, it is interesting to note that destabilization of only the ground (or the excited state) by mutagenesis is expected to increase k_1 (respectively k_{-1}) and therefore the global exchange rate k_{ex} , as experimentally observed.”

Supporting Figure 9: Schematic diagrams of the energy landscapes for the exchange between the ATP-lid open (ground) and closed (excited) states of WT, R60A, and R46A variants of HSP90 α -NTD. Kinetic exchange rates at 288 K are displayed both for the WT protein and the mutants. Assuming, that the mutations do not influence the energy of the transition state and destabilize only the state for which the segment [98-136] is in close proximity to the mutation, we can deduce that the R46A (R60A) mutation leads to an increase of k_1 (respectively k_{-1}), and therefore of the global exchange constant $k_{ex} = k_1 + k_{-1}$ as observed experimentally.

In addition, there is currently no experimental evidence showing that the excited state detected by the CPMG RD is indeed the open state of the protein. As the exchange determined by CPMG RD is quite fast for such large conformational changes, this point must be further validated. The authors can compare the chemical shifts of the excited state to see if these indeed resemble the close state conformation.

We thought to use chemical shift differences to validate the model, but we do not observe directly the chemical shifts of the closed excited state. Furthermore, the error of predicted chemical shifts for methyl groups from protein structures using available softwares (Han et al. J. Biomol NMR 2011 – doi: 10.1007/s10858-011-9478-4; Sahakyon et al. J. Biomol NMR 2011 – doi : 10.1007/s10858-011-9524-2) (1.2 ppm for ^{13}C), combined with the structure uncertainties of the excited closed state (rmsd 2.08 Å for heavy atoms of ATP-lid) precludes the use of chemical shifts to identify the closed conformation. As an example, the figure below represents the ^{13}C -methyl chemical shifts predicted vs the experimentally observed chemical shifts for the ground open state for which the structure is known with a higher precision compared to the structure of the excited closed state.

Figure legend: Predicted ^{13}C values obtained with SHIFTX2 (Han et al. J. Biomol NMR 2011 – doi: 10.1007/s10858-011-9478-4) for the open ATP-lid structure plotted as a function of the experimental ^{13}C values obtained from NMR experiments for HSP90-NTD-WT methyl groups. Each methyl group resonance has been corrected with the average chemical shift of methyl groups of the same type (*i.e.* Ala- β , Ile- δ_1 , Leu- δ_2 , Met- ϵ , Thr- γ , Val- γ_2). In red, blue, orange, green, magenta and black are represented the resonances of the alanine, isoleucine, leucine, threonine, methionine and valine methyl groups, respectively. The black line represents a linear correlation ($x = y$).

Given the uncertainties on predicted chemical shifts and the lower precision of the structure of the excited closed state, such analysis is risky and prone to misinterpretation, therefore, we have preferred not to discuss comparison of chemical shifts in the article in order to avoid any risk of overinterpretation of our data. Nevertheless, it is interesting to note that the population of excited state derived from CPMG RD analysis (3.2%) is in good agreement with the population estimated from our NOE-derived closed state structure (5%) using online PDBcor statistical analysis tool (www.pdbcor.ethz.ch), as indicated at the end of the Results section (page 15, lines 9-12):

“It is interesting to note, that the population of the excited state derived from CPMG data analysis is in good agreement with the 5% population estimated for the closed state using a statistical analysis of our NOE-derived solution structure ensembles using PDBcor⁵⁶.”

2) This study was performed with an isolated NTD domain, and there are no experiments tying these findings to either the structure or the function of the full length Hsp90 protein. In my opinion, such experiments are crucial to prove the biological importance of the newly found closed state of the NTD. Likewise, the manuscript states that the structure of the closed state precludes ATP binding, however this statement does not appear to be supported by experimental evidence.

Despite our efforts, full length dimeric human HSP90 α does not give rise to useful NMR spectra hampering quantitative analysis (see answer to reviewer 2). Nevertheless, to reinforce the biological importance of our results we have added new data in presence of a nucleotide and a ligand. We have demonstrated that the sampling of the excited state is preserved in presence of these ligands (see answer to reviewer 2). We agree with the reviewer that our data do not allow to draw conclusions regarding the binding of ATP to either the open or closed states, such conclusions are absent in the revised manuscript.

Minor comments-

1) All the experiments in this study were performed without a nucleotide. Can the authors speculate how the results would change if ATP or ADP were included in their measurements?

We are now presenting new NOESY data acquired in presence of ADP and a resorcinol inhibitor and we have shown that similar results are obtained upon addition of these ligands. See answer to reviewer 2 for detailed information

2) In the methods sections it is stated that both methyl-carbon and nitrogen CPMG experiments were recorded, however no data for the nitrogen experiments is shown.

Initial experiments have shown that both methyl and nitrogen CPMG experiments provide the same information, but methyl edited NMR experiments are characterized by a higher signal to noise ratio. Therefore, most of the experiments were acquired using methyl labelled samples. In the revised version, we have added a supporting figure presenting ^{15}N relaxation dispersion profiles acquired at 3 magnetic fields on WT-HSP90 α -NTD.

Supporting Figure S8B: Global fitting of ^{15}N relaxation dispersion data of HSP90 α -NTD-WT obtained at 293 K at three different magnetic field strengths. Graphs representing experimental R_2^{eff} as a function of the CPMG frequency (ν_{CPMG}) obtained for four residues: T36, A101, S129 and E158 out of 14 residues fitted globally (global fit represented by the black curves). The conformational exchange rate and population of the excited state obtained with a global fit of ^{15}N relaxation dispersion experiments are $2435 \pm 89 \text{ s}^{-1}$ and $2.5 \pm 0.4 \%$, respectively.

3) In the discussion section it is stated that the R46A mutant of Hsp90 stays monomeric in solution, however no experimental evidence (such as SEC-MALS for example) is provided.

We thank the reviewer for this suggestion. We have performed SEC-MALS analysis of both WT- and R46A-HSP90 α -NTD. The profiles, very similar and characteristic of monomeric proteins, are provided as Supporting Fig. 12 in the revised version. We added the following sentence in Discussion section (page 18, lines 8-10):

“SEC-MALS analysis of both WT and R46A mutant of HSP90 α –NTD confirm that the protein is monomeric in solution (Supporting Fig. 12).”

Supporting Figure 12: SEC-MALS analysis of uncleaved R46A-HSP90 α -NTD (A) and WT-HSP90 α -NTD (B) using gel filtration column Superdex 75 PG (GE Healthcare) equilibrated with NMR buffer.

Reviewers' Comments:

Reviewer #1:

Remarks to the Author:

The authors have adequately addressed all concerns raised. I suggest accepting of this outstanding work.

Reviewer #2:

Remarks to the Author:

The authors have answered all my comments in a satisfactory manner and have updated the manuscript accordingly. As far as I understand they also have done so with the comments from the other reviewers. Consequently, I strongly recommend the publication of the manuscript in the current, revised state.

Reviewer #3:

Remarks to the Author:

The authors have performed several additional experiments that address the majority of the reviewers' comments and greatly improve the quality of the manuscript.

There is still one point, however, that in my opinion must be addressed prior to publication.

The authors determined the structure of the transient (~5% populated) Hsp90 NTD closed state using a set of NOE distance restraints. This part of the manuscript, in my opinion, is highly supported by experimental evidence and further validated by specific mutations that destabilize either the ground (open) or the excited (closed) states.

The authors also measured CPMG RD NMR experiments and found that the ground (open) state exchanges at $\sim 2400 \text{ s}^{-1}$ rate with an excited state. However, no experimental evidence is provided to show that the excited state detected by the CPMG RD is indeed the closed state of the NTD.

This is particularly problematic as the excited state (from CPMG) has a lifetime of 0.3 ms, which is, in principle, too short to allow detection of NOE cross-peaks.

Furthermore, the authors performed additional experiments with a mutant that destabilizes the closed NTD state. This mutant, despite showing no NOEs for the closed state, still showed CPMG dispersions. It is of course possible that the population of the closed state is too low for detection of NOEs while still showing dispersions. In that case, though, the size of the dispersions should be significantly smaller compared to those of the WT protein. Was such behavior observed? If not, then that would preclude the closed state from being the one observed in the CPMG experiments, and the conclusions regarding this should be amended accordingly. In either case, the comparison of dispersions for the same residues for WT and the two mutants should be added as a supplementary figure to the manuscript.

I must point out that the above should not prevent the manuscript from being published. Both structural data and dynamics measurements are of high quality on their own, however it is just not clear whether the two measurements indeed report on the same process and the same excited (closed) state.

My recommendation would be to either provide additional experimental evidence linking the two data sets, or alternatively to scale back the claims regarding the thermodynamic and kinetics

measurements for the closed state of the NTD.

Reviewer #4:

Remarks to the Author:

The authors of the manuscript have performed a very thorough and comprehensive revision, adding a significant amount of new data and improving the overall presentation. The conclusions of the manuscript and its novel aspects are better substantiated in the revision.

Manuscript NCOMMS- 22-12045A
Revised version
Point-by-point response to the reviewers' comments

Colour code:

Black - Reviewers' remarks

Green - authors' comment

red- updated parts of the manuscript

Gray - non-modified parts of the manuscript

Reviewer #1 (Remarks to the Author):

The authors have adequately addressed all concerns raised. I suggest accepting of this outstanding work.

We thank the reviewer #1 for the positive comments.

Reviewer #2 (Remarks to the Author):

The authors have answered all my comments in a satisfactory manner and have updated the manuscript accordingly. As far as I understand they also have done so with the comments from the other reviewers. Consequently, I strongly recommend the publication of the manuscript in the current, revised state.

We thank the reviewer #2 for the positive comments.

Reviewer #3 (Remarks to the Author):

The authors have performed several additional experiments that address the majority of the reviewers' comments and greatly improve the quality of the manuscript.

There is still one point, however, that in my opinion must be addressed prior to publication.

The authors determined the structure of the transient (~5% populated) Hsp90 NTD closed state using a set of NOE distance restraints. This part of the manuscript, in my opinion, is highly supported by experimental evidence and further validated by specific mutations that destabilize either the ground (open) or the excited (closed) states.

The authors also measured CPMG RD NMR experiments and found that the ground (open) state exchanges at ~2400 s⁻¹ rate with an excited state. However, no experimental evidence is provided to show that the excited state detected by the CPMG RD is indeed the closed state of the NTD.

This is particularly problematic as the excited state (from CPMG) has a lifetime of 0.3 ms, which is, in principle, too short to allow detection of NOE cross-peaks.

Furthermore, the authors performed additional experiments with a mutant that destabilizes the closed NTD state. This mutant, despite showing no NOEs for the closed state, still showed CPMG dispersions. It is of course possible that the population of the closed state is too low for detection of NOEs while still showing dispersions. In that case, though, the size of the dispersions should be significantly smaller compared to those of the WT protein. Was such behavior observed? If not, then that would preclude the closed state from being the one observed in the CPMG experiments, and the conclusions regarding this should be amended accordingly. In either case, the comparison of dispersions for the same residues for WT and the two mutants should be added as a supplementary figure to the manuscript.

I must point out that the above should not prevent the manuscript from being published. Both structural data and dynamics measurements are of high quality on their own, however it is just not clear whether the two measurements indeed report on the same process and the same excited (closed) state.

My recommendation would be to either provide additional experimental evidence linking the two data sets, or alternatively to scale back the claims regarding the thermodynamic and kinetics measurements for the closed state of the NTD.

We thank the reviewer #3 for the positive comments. As suggested, by reviewer 3 we have scale back our claims regarding interpretation of our relaxation-dispersion data and we have added the following sentences:

Results section (page 15, lines 8-12): While there is no direct experimental evidence that the excited state detected by the CPMG relaxation-dispersion is indeed the HSP90 α -NTD closed state structure determined combining ¹³CH₃-NOESY and mutagenesis, the simplest model explaining all the experimental data is the two-state model where the excited state detected by the CPMG relaxation-dispersion is indeed the HSP90 α -NTD closed state (Fig. 3b).

Discussion section (p17, lines 5-7): The transition pathway between open and closed ATP-lid states, distant by up to 30 Å, is obviously more complicated than a simple two-state model used here for the quantitative analysis of our ~~NMR data~~ CPMG relaxation-dispersion data and we cannot exclude that intermediate states could also be involved in the broadening of NMR signals.

Reviewer #4 (Remarks to the Author):

The authors of the manuscript have performed a very thorough and comprehensive revision, adding a significant amount of new data and improving the overall presentation. The conclusions of the manuscript and its novel aspects are better substantiated in the revision.

We thank the reviewer #4 for the positive comments.